# CoLA: Cross-Modal Low-rank Adaptation for Multimodal Downstream Tasks

**Wish Suharitdamrong** [1]  **Tony Alex** [1 2]  **Muhammad Awais** [1 2]  **Sara Atito** [1 2]

## Abstract

Foundation models have revolutionized AI, but adapting them efficiently for multimodal tasks, particularly in dual-stream architectures composed of unimodal encoders, such as DINO and BERT, remains a significant challenge. Parameter-Efficient Fine-Tuning (PEFT) methods like Low-Rank Adaptation (LoRA) enable lightweight adaptation, yet they operate in isolation within each modality, limiting their ability in capturing cross-modal interactions. In this paper, we take a step in bridging this gap with Cross-Modal Low-Rank Adaptation (CoLA), a novel PEFT framework that extends LoRA by introducing a dedicated inter-modal adaptation pathway alongside the standard intra-modal one. This dual-path design enables CoLA to adapt unimodal foundation models to multimodal tasks effectively, without interference between modality-specific and cross-modal learning. We evaluate CoLA across a range of vision-language (RefCOCO, RefCOCO+, RefCOCOg) and audio-visual (AVE, AVS) benchmarks, where it consistently outperforms LORA, achieving a relative gain of around 3% and 2%, respectively, while maintaining parameter efficiency. Notably, CoLA enables the first multitask PEFT framework for visual grounding, bridging a key gap in efficient multimodal adaptation. Code is available at https://github.com/peterwisu/CoLA

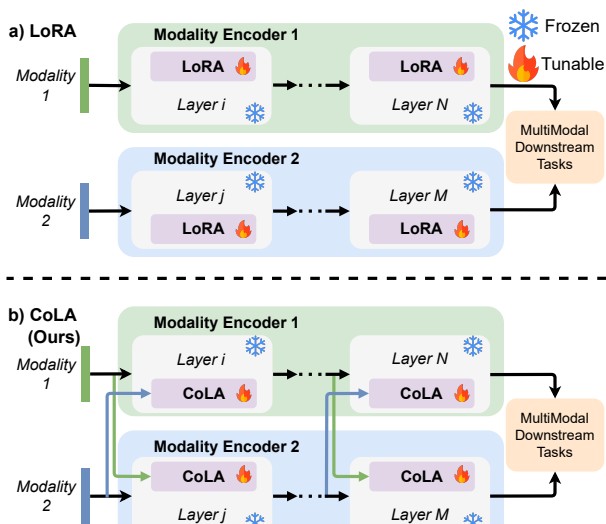

*Figure 1.* Comparison of LoRA and CoLA in dual-encoder architectures for multimodal tasks. (a) LoRA applies independent low-rank adaptation within each modality without cross-modal interaction. (b) CoLA enables cross-modal interaction through inter-modal fusion pathways, allowing information exchange between Modality 1 and Modality 2 during the low-rank adaptation process. Modality 1 and Modality 2 can be vision, language, or audio. The multimodal tasks include vision-language (REC and RES) and audio-visual (AVE and AVS) downstream tasks.

## 1. Introduction

The widespread usage of foundation models (Devlin et al., 2019; Radford et al., 2021; Girdhar et al., 2023; Oquab

et al., 2023; Elizalde et al., 2023) has demonstrated their ability to generalize across various downstream tasks in both unimodal and multimodal domains. However, as foundation models continue to grow in scale, performing full fine-tuning becomes increasingly costly and computationally impractical. Parameter-efficient fine-tuning (PEFT) has been introduced to mitigate this issue, which aim to adapt large pre-trained models using a small fraction of trainable parameters. Among PEFT methods (Houlsby et al., 2019; Hu et al., 2022; Lester et al., 2021; Li & Liang, 2021), Low-Rank Adaptation (LoRA) (Hu et al., 2022) has emerged as a particularly popular approach due to its simplicity and effectiveness through its low-rank structure.

The representation from unimodal pre-trained encoders can be highly effective in a dual-encoder architecture setting for multimodal downstream tasks. PEFT methods such as

[1]Surrey Institute for People-Centred AI, University of Surrey, Guildford, GU2 7XH, UK [2]Centre for Vision, Speech and Signal Processing (CVSSP), University of Surrey, UK. Correspondence to: Wish Suharitdamrong <ws00372@surrey.ac.uk>.

*Proceedings of the $43^{rd}$ International Conference on Machine Learning*, Seoul, South Korea. PMLR 306, 2026. Copyright 2026 by the author(s).

LoRA can be applied to these architectures for multimodal tasks, as shown in Figure 1. However, the adaptation from LoRA is modality-specific and lacks cross-modal awareness, limiting the opportunity to leverage complementary information between modalities. Prior works (Yang et al., 2022; Ye et al., 2022; Zhang et al., 2022; Deng et al., 2023; Su et al., 2023b;a; Yao et al., 2024) have addressed modality-specific features in the unimodal backbone by enabling cross-modal interaction through their intermediate layers. This provides cross-modal awareness to their extracted representations for multimodal tasks. Building on these insights, the integration of cross-modal awareness to LoRA would be beneficial to enhance the adaptation process, improving performance in multimodal downstream tasks.

These limitations highlight a gap in applying LoRA to dual-encoder architectures, where cross-modal awareness is essential for effective multimodal adaptation of unimodal foundation models. We introduce CoLA (Cross-modal Low-rank Adaptation), which provides both intra-modal adaptation and inter-modal fusion pathways for effective cross-modal adaptation in dual-encoder architectures. While LoRA provides efficient adaptation in the intra-modal pathway, CoLA extends its formulation with the inter-modal low-rank pathway, constructing fusion weights generated by cross-modal features. This enables the efficient fine-tuning process to handle both intra- and inter-modal information efficiently, while maintaining clean separation between modality-specific and cross-modal computations. With CoLA, bidirectional cross-modal interaction can occur at any linear component of the modules, enabling symmetric fusion between both modalities. The illustration of CoLA integrated into a dual-encoder architecture, compared with LoRA, is shown in Figure 1. The summary of our contributions is listed as follows:

- We present CoLA, which extends the capability of LoRA with the integration of cross-modal awareness, improving the performance of dual-encoder architectures for multimodal tasks.

- Experimental results demonstrate the effectiveness of CoLA across multiple multimodal downstream tasks, showing consistent improvements over existing PEFT.

- Comprehensive experiments and ablation studies validate the design choices and effectiveness of CoLA's components, analyzing the contributions of intra-modal adaptation and inter-modal fusion pathways.

## 2. Background

### 2.1. Parameter-Efficient Fine-Tuning (PEFT)

PEFT aims to enable efficient adaptation by updating only a small fraction of parameters. These approaches include adapter methods (Houlsby et al., 2019) introducing small trainable modules, prompt-based strategies (Lester et al., 2021; Li & Liang, 2021) optimizing input representations, and low-rank methods (Karimi Mahabadi et al., 2021; Hu et al., 2022) that reparameterize model weights through low-rank decomposition. LoRA (Hu et al., 2022) has become the most widely adopted method, using the product of two low-rank matrices for efficient adaptation without inference overhead. Recent PEFT approaches for dual-encoder architectures in multimodal tasks (Xu et al., 2023; Lin et al., 2023; Duan et al., 2023; Wang et al., 2024b; Xiao et al., 2024; Wang et al., 2024a; Shi et al., 2025; Huang et al., 2025) have enabled cross-modal interaction through adapter modules applied sequentially or in parallel to frozen backbones. However, these methods typically fuse cross-modal information at the module level and are often designed for specific modality pairs or downstream tasks. In contrast, CoLA facilitates cross-modal interaction within individual linear components and can be applied to any combination of modalities or downstream tasks.

### 2.2. Unimodal Foundation Model for Multimodal Tasks

CLIP (Radford et al., 2021) and other jointly trained multimodal encoders (Girdhar et al., 2023; Elizalde et al., 2023) may discard task-relevant information by prioritizing alignment over modality-specific representations and may not have architectures well-suited for specific downstream tasks. This motivates the use of unimodal foundation models, such as (Siméoni et al., 2025; Atito et al., 2026) for vision and (Chen et al., 2023; Alex et al., 2025) for audio, in a dual-encoder architecture setting. In the vision-language domain, DETRIS (Huang et al., 2025) replaces CLIP's vision encoder with DINOv2 (Oquab et al., 2023) while pairing it with CLIP's text encoder, leveraging the strong generalization of self-supervised learning to address CLIP's limitations in fine-grained spatial understanding. Other works (Ye et al., 2022; Yang et al., 2022; Deng et al., 2023; Zhang et al., 2022; Su et al., 2023a; Yao et al., 2024; Su et al., 2023b) also employ separate unimodal encoders for vision-language tasks, which are better suited for their downstream tasks, demonstrating the practical viability of this approach. In the audio-visual domain, LAVisH (Lin et al., 2023) and STG-CMA (Wang et al., 2024a) utilize a pre-trained vision model, sharing its weights for both visual and audio modalities, leveraging the transferability of visual representations to audio features through PEFT modules. On the other hand, DG-SCT (Duan et al., 2023) employs separate unimodal encoders for audio-visual modalities, leveraging the strong modality-specific representations from vision and audio foundation models. These studies show the power of unimodal foundation models for multimodal tasks.

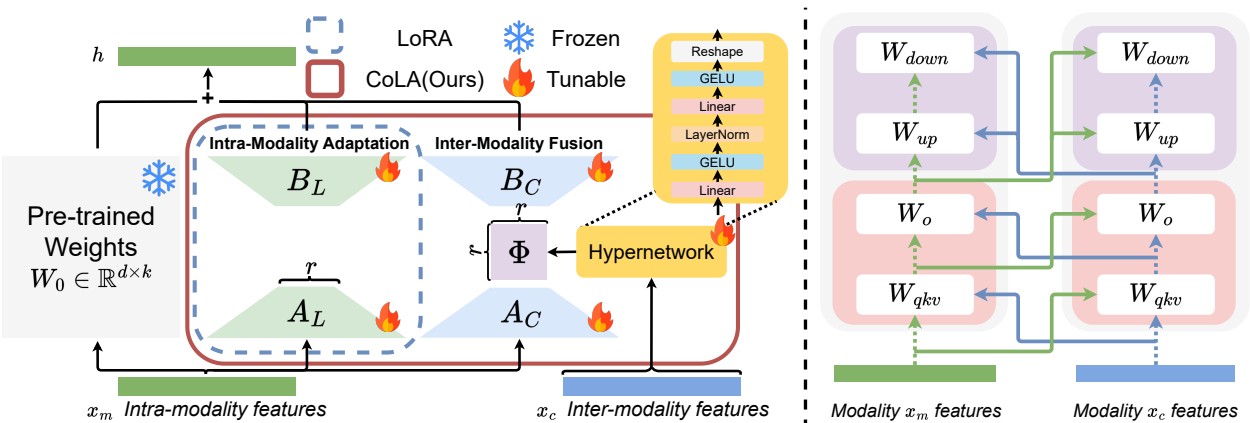

*Figure 2.* (Left) The overall architecture of CoLA applied to pre-trained linear components $W_0$ in transformer blocks with the intra-modal pathway $\Delta W_L$ and inter-modal fusion pathway $\Delta W_C$ in Equation 4, which integrates dynamic weights from cross-modal features via a hypernetwork. (Right) Illustration of the progressive cross-modal propagation between dual encoders, transferring cross-modal features to linear component with CoLA in self-attention (SA: $W_{qkv}$), output projection (OUT: $W_o$) and FFN module up-projection (UP: $W_{up}$), and down-projection (DOWN: $W_{down}$).

## 3. Method

In this section, we first outline how LoRA operates within transformer architectures. Building on this foundation, we introduce Cross-Modal Low-Rank Adaptation (CoLA), a novel extension designed to enable cross-modal interactions in dual-stream multimodal settings.

### 3.1. LoRA in Transformer Architectures

In the Transformer encoder architecture, each encoder layer generally consists of two main modules: Multi-Head Self-Attention (MHSA) and a feed-forward network (FFN). The MHSA consists of several linear projection matrices $W_q, W_k \in \mathbb{R}^{d_k \times d_{model}}$, $W_v \in \mathbb{R}^{d_v \times d_{model}}$ and $W_o \in \mathbb{R}^{d_{model} \times d_v}$ to capture inter-token relationships and contextual dependencies across the token's sequence $x \in \mathbb{R}^{N \times d_{model}}$. The mathematical formulation of MHSA is given in equation (1).

$$\text{MHSA}(X) = W_o \left[ (W_v X) \sigma \left( \frac{(W_k X)^T (W_q X)}{\sqrt{d_k}} \right) \right] \quad (1)$$

where $\sigma(\cdot)$ is the softmax function, $N$ is the number of tokens, and $d_k$, $d_v$, $d_{model}$ denote the query, key, value, and model dimensions, respectively. For simplicity, we skip the layer normalization and residual connection and assume a single attention head. The FFN module consists of two linear layers $W_{up} \in \mathbb{R}^{d_{ffn} \times d_{model}}$ and $W_{down} \in \mathbb{R}^{d_{model} \times d_{ffn}}$ with $\phi(\cdot)$ non-linear activation function to apply non-linear transformations to each token representation. Here, $d_{ffn}$ is the feed-forward hidden dimension, typically $4 \times d_{model}$. The FFN computation can be formulated as shown in equation (2).

$$\text{FFN}(x) = W_{\text{down}} \phi(W_{\text{up}} x) \quad (2)$$

Similarly, for simplicity, we skip normalization, residual connection, and their bias term in this formulation. LoRA can be applied individually to any linear component in these modules, where $W_0 \in \mathbb{R}^{d_{out} \times d_{in}}$ denotes its original pre-trained weight matrix, which remains fixed during adaptation. Here, $d_{out}$ and $d_{in}$ represent the output and input dimensions, respectively. LoRA approximates the weight update $\Delta W_L \in \mathbb{R}^{d_{out} \times d_{in}}$ by decomposing its into smaller low-rank matrices where $B_L \in \mathbb{R}^{d_{out} \times r}$ and $A_L \in \mathbb{R}^{r \times d_{in}}$ are trainable low-rank matrices with the rank $r << min(d_{out}, d_{in})$, significantly reducing the number of trainable parameters in the adapation process. This low-rank adaptation can be expressed as shown in equation (3).

$$h = W_0 x + \Delta W_L x = W_0 x + \frac{\alpha}{r} B_L A_L x \quad (3)$$

where $h \in \mathbb{R}^{N \times d_{out}}$ and $x \in \mathbb{R}^{N \times d_{in}}$ are the output and input, respectively. $\alpha$ is a scaling factor controlling the magnitude of $\Delta W_L$, with effective updates scaled by $\frac{\alpha}{r}$. Matrix $A_L$ is initialized from uniform Kaiming (He et al., 2015) while $B_L$ is zero-initialized, ensuring $\Delta W_L = B_L A_L$ starts at zero to begin from pre-trained knowledge without low-rank component interference.

### 3.2. Proposed Cross-Modal LoRA (CoLA)

From LoRA, we have $W_0 + \Delta W_L$, where this refers to the intra-modal adaptation. As we discussed earlier, our motivation is to extend LoRA to multimodal settings by simply adding a inter-modal fusion pathway $\Delta W_C \in \mathbb{R}^{d_{out} \times d_{in}}$ to LoRA in equation (3) for cross-modal interaction as shown in equation (4).

$$h_m = W_0^m x_m + \underbrace{\Delta W_L^m x_m}_{\text{intra-modal}} + \underbrace{\Delta W_C^m x_m}_{\text{inter-modal}} \quad (4)$$

where $m$ denotes modality (e.g., vision, audio), $x_m \in \mathbb{R}^{N_m \times d_m}$ is the input with $N$ tokens and $d$ feature dimensions ($d_m = d_{in}$), and $\Delta W_L^m$ is the intra-modal adaptation weight from LoRA in equation (3), as illustrated in Figure 2 (Left). In this section, we discuss how $\Delta W_C^m$ is obtained and how cross-modal features are propagated through the dual-encoder architecture. First, the added inter-modal weight $\Delta W_C^m \in \mathbb{R}^{d_{out} \times d_{in}}$ can be decomposed into low-rank matrices $B_C^m \in \mathbb{R}^{d_{out} \times r}$ and $A_C^m \in \mathbb{R}^{r \times d_{in}}$ with initialization similar to LoRA. For simplicity, we use the same rank $r$ for both LoRA and CoLA pathways. To incorporate cross-modal dependencies, we introduce a square matrix $\Phi^m \in \mathbb{R}^{r \times r}$ as an intermediate transformation matrix between $B_C^m$ and $A_C^m$. Additionally, we utilize a learnable scalar $\lambda$ to control the contribution of $\Delta W_C^m$, unlike intra-modal adaptation, which uses a static scaling factor as formulated in equation (5) and shown in the inter-modality fusion pathway of Figure 2 (Left).

$$\Delta W_C^m = \lambda B_C^m \Phi^m A_C^m \tag{5}$$

The $\Phi^m$ matrix is dynamically generated from cross-modal features $x_c \in \mathbb{R}^{N_c \times d_c}$ from modality $c$ of the paired encoder via a hypernetwork, as depicted in Figure 2. This allows the inter-modal adaptation of modality $m$ with cross-modal information from modality $c$. To obtain $\Phi^m$, we first extract a global representation $\bar{x}_c$ by either averaging $x_c$ along the token dimension $N_c$ or utilizing the [CLS] token, depending on the model architecture and downstream task. We then pass it into a hypernetwork consisting of two linear layers with a non-linear function $\phi(\cdot)$ and layer normalization $\text{LN}(\cdot)$, as shown in equation (6).

$$\Phi^m = \text{LN}(W_{up}^m \text{LN}(\phi(W_{down}^m \bar{x}_c))) \tag{6}$$

where $W_{down}^m \in \mathbb{R}^{\frac{d_c}{\gamma} \times d_c}$ and $W_{up}^m \in \mathbb{R}^{r^2 \times \frac{d_c}{\gamma}}$ are the weight matrices, $\gamma$ is reduction factor and $r$ is rank of low-rank matrices. The hypernetwork projects $\bar{x}_c$ into an $r^2$-dimensional space, reshaped into the $r \times r$ matrix $\Phi^m$. The final CoLA can be formulated as the composition of two distinct low-rank pathways, $\Delta W_L$ for intra-modal adaptation and $\Delta W_C$ for inter-modal fusion, as shown in Figure 2 (Left). Note that when adapting modality $c$, each modality has its own pre-trained $W_0$. For example, The pre-trained weight $W_0^c$ of modality $c$ would have its own adaptation weights $W_L^c$ and $W_C^c$. The formulation follows the same dual-pathway structure where $\Delta W_C^c$ uses features $x_m$ from modality $m$ to generate $\Phi^c$ via the hypernetwork, enabling symmetric cross-modal adaptation.

Having established how CoLA computes cross-modal adaptations, we now describe how these adaptations are integrated into the dual-encoder architecture. Cross-modal features are progressively propagated through the dual-encoders, where CoLA is applied to linear layers, as illustrated in Figure 2 (Right). Features from each encoder are

updated and passed to CoLA in the paired encoder, evolving as they flow through self-attention, output projection, and FFN layers, as shown in Algorithm 1.

**Algorithm 1** PyTorch-style pseudocode for dual encoder forward pass with CoLA. SA: Self-Attention, WO: Out-Projection, FFN: Feed-Forward Network

```
1: def forward(self, x_m, x_c):
2:     """
3:     x_m:  modality m input features
4:     x_c:  modality c input features
5:     """
6:     for layer in self.layers:
7:         # Self-Attention stage (W_q, W_k, W_v)
8:         a_m = layer.SA_m(x_m, x_c)
9:         a_c = layer.SA_c(x_c, x_m)
10:        # Attention Output stage (W_o)
11:        o_m = layer.WO_m(a_m, a_c) + x_m
12:        o_c = layer.WO_c(a_c, a_m) + x_c
13:        # Feed-Forward Network stage (W_up, W_down)
14:        x_m = layer.FFN_m(o_m, o_c) + o_m
15:        x_c = layer.FFN_c(o_c, o_m) + o_c
16:    return x_m, x_c
```

### 3.3. Theoretical Analysis of CoLA

Having defined CoLA's two pathways, we now analyse what the dynamic, input-dependent $\Phi$ contributes beyond LoRA's static update. We focus on three properties of the inter-modal weight $\Delta W_C = \lambda B_C \Phi A_C$: the rank of a single update, the dimensionality of the effective weight space across inputs, and the gradient structure of the two pathways. For readability we suppress the modality index $m$, since the analysis is identical for each modality.

**Rank of the update.** The factors $B_C \in \mathbb{R}^{d_{out} \times r}$ and $A_C \in \mathbb{R}^{r \times d_{in}}$ are low rank with $r \ll \min(d_{in}, d_{out})$, while $\Phi \in \mathbb{R}^{r \times r}$. By the property of rank under matrix multiplication, $\text{Rank}(XY) \leq \min(\text{Rank}(X), \text{Rank}(Y))$, the product $B_C \Phi A_C$ satisfies

$$\text{Rank}(\Delta W_C) \leq \min\big(\text{Rank}(B_C), \text{Rank}(\Phi), \\ \text{Rank}(A_C), r\big). \tag{7}$$

Since $\text{Rank}(B_C), \text{Rank}(A_C) \leq r$, we have $\text{Rank}(\Delta W_C) \leq r$ regardless of $\text{Rank}(\Phi)$. A full-rank $\Phi$ can preserve the rank-$r$ capacity when $B_C$ and $A_C$ also have rank $r$, while a rank-deficient $\Phi$ reduces it further, acting as implicit gating on cross-modal interaction. Thus $\Delta W_C$ matches LoRA's rank-$r$ limit.

**Expressive power across inputs.** The rank bound above holds for any single $\Delta W_C$. Across multiple cross-modal inputs, however, $\Phi$ varies with the input. Let $\Phi^{(k)}$ denote

the value of $\Phi$ produced by the $k$-th cross-modal input, and let $\Delta W_C^{(k)} = \lambda B_C \Phi^{(k)} A_C$. Vectorising both sides using the identity $\text{Vec}(XYZ) = (Z^\top \otimes X)\text{Vec}(Y)$ gives

$$\text{Vec}\big(\Delta W_C^{(k)}\big) = \lambda \,(A_C^\top \otimes B_C)\,\text{Vec}(\Phi^{(k)}). \qquad (8)$$

The matrix $A_C^\top \otimes B_C$ does not depend on $k$, so every $\text{Vec}(\Delta W_C^{(k)})$ lies in the image of this fixed linear map. By the Kronecker rank identity $\text{Rank}(X \otimes Y) = \text{Rank}(X)\,\text{Rank}(Y)$,

$$\text{Rank}(A_C^\top \otimes B_C) = \text{Rank}(A_C)\,\text{Rank}(B_C) \leq r^2. \qquad (9)$$

This yields the following result.

**Theorem 3.1** (Expressive power of CoLA). *Across $K$ cross-modal inputs producing updates $\{\Delta W_C^{(k)}\}_{k=1}^K$, the linear span $\text{span}\big\{Vec(\Delta W_C^{(k)})\big\}_{k=1}^K$ has dimension at most $\min(K, r^2, d_{out}d_{in})$.*

LoRA's update $\Delta W = BA$ is fixed across inputs: it has rank $r$ in activation space and spans a single direction in weight space. CoLA's updates also have rank at most $r$ per input (by Eq. 7), but span up to $r^2$ directions in weight space across inputs. When $\Phi$ varies across inputs, the dynamic pathway expands the family of representable input-dependent weight updates compared with a fixed LoRA update.

**Gradient decoupling.** Let $\delta = \partial\mathcal{L}/\partial h$ denote the upstream gradient at the layer output $h$. The parameter gradients for the intra-modal pathway are

$$\frac{\partial\mathcal{L}}{\partial B_L} = \frac{\alpha}{r}\,\delta\,(A_L x)^\top,$$
$$\frac{\partial\mathcal{L}}{\partial A_L} = \frac{\alpha}{r}\,B_L^\top\,\delta\,x^\top, \qquad (10)$$

and for the inter-modal pathway are

$$\frac{\partial\mathcal{L}}{\partial B_C} = \lambda\,\delta\,(\Phi A_C x)^\top,$$
$$\frac{\partial\mathcal{L}}{\partial \Phi} = \lambda\,B_C^\top \delta\,(A_C x)^\top,$$
$$\frac{\partial\mathcal{L}}{\partial A_C} = \lambda\,\Phi^\top B_C^\top \delta\,x^\top. \qquad (11)$$

Each pathway's gradients depend only on its own factors and the shared upstream signal $\delta$, so the two pathways update through disjoint parameter sets. This motivates the non-shared design of the low-rank matrices between the two pathways, which is confirmed empirically in Sec. 5.1.

# 4. Experiments & Results

We conduct several experiments to demonstrate the effectiveness of CoLA on multimodal tasks. We evaluate CoLA

on referring expression comprehension (REC) and referring expression segmentation (RES) for vision-language tasks, and audio-visual event localization (AVE) and audio-visual segmentation (AVS) for audio-visual tasks. First, we compare CoLA with LoRA to isolate the contribution of our cross-modal mechanism. We then compare CoLA with existing dual-encoder PEFT methods designed for specific multimodal tasks to demonstrate its effectiveness in the broader context of multimodal adaptation methods. Below, we provide brief implementation details, task descriptions, and dataset information for these experiments. More comprehensive details can be found in the Appendix A.

## 4.1. Multimodal Tasks & Experimental Setup

### 4.1.1. VISION-LANGUAGE TASKS

Both REC and RES involve grounding language expressions to visual objects through bounding box localization and pixel-level segmentation, respectively. We utilized common referring expression datasets RefCOCO (Yu et al., 2016), RefCOCO+ (Yu et al., 2016), and RefCOCOg dataset (Mao et al., 2016), derived from MSCOCO (Lin et al., 2014), which provide both annotations for both tasks. See Appendix B.1 for more details. For REC, we use accuracy to measure predicted bounding boxes with an IoU greater than 0.5 against the ground truth. For RES, we use the mean IoU (mIoU), which measures the average IoU between the predicted and ground truth masks. For implementation, we utilize ViT-B (Dosovitskiy et al., 2020), pre-trained on MSCOCO, with adaptations introduced from ViTDet (Li et al., 2022) and BERT-B (Devlin et al., 2019) for vision and language backbones, respectively. For the multimodal task decoder module, we utilize multi-task visual grounding decoder from EEVG (Chen et al., 2024) to perform both REC and RES simultaneously. We freeze the vision and language backbone parameters and keep the multi-task decoder trainable. We applied CoLA to all Q, K, V, and FFN components of both backbones. We use the rank of 16 for both intra- and inter-modal pathways in CoLA. For training details, refer to Appendix A.1.

### 4.1.2. AUDIO-VISUAL TASKS

AVE focuses on recognizing audio-visual events that are visible and audible throughout temporal segments in videos. In contrast, AVS segments objects that generate sound during the corresponding image frame. We utilized the AVE dataset (Tian et al., 2018) for AVE and the AVSBench-S4 dataset (Zhou et al., 2022) for AVS (See Appendix B.2 for more details), using accuracy and mIoU score as the evaluation metrics for these tasks, respectively. For implementation on AVE, we utilized DINOv2-B-14 (Oquab et al., 2023) as the vision backbone, with SSLAM (Alex et al., 2025) as the audio backbone. The obtained features are concatenated

*Table 1.* Comparison of CoLA and LoRA on vision-language tasks with rank-matched (r=16) and parameter-matched (r=54) baselines. CoLA outperforms both LoRA configurations on both REC and RES tasks, demonstrating performance gains from cross-modal integration rather than parameter increase. Parameter Update and Total Parameters represent parameter counts measured in millions. The Update Ratio column indicates the percentage of trainable parameters relative to total parameters in the whole model.

| Method | Param Update | Total Param | Update Ratio | RefCOCO | | | RefCOCO+ | | | RefCOCOg | | Avg ↑ |
|---|---|---|---|---|---|---|---|---|---|---|---|---|
| | | | | val | testA | testB | val | testA | testB | val | test | |
| *Referring Expression Comprehension (REC)* | | | | | | | | | | | | |
| LoRA(r=16) | 28.0 | 223.4 | 12.5% | 88.7 | 90.5 | 86.0 | 78.5 | 83.3 | 70.6 | 80.2 | 80.2 | 82.3 |
| LoRA(r=54) | 40.6 | 236.0 | 17.2% | 88.4 | 90.2 | 85.9 | 78.3 | 82.9 | 69.6 | 79.7 | 79.3 | 81.8 |
| CoLA(r=16) | 40.5 | 236.0 | 17.2% | **89.4** | **91.0** | **86.9** | **79.6** | **84.7** | **71.9** | **81.7** | **81.8** | **83.4** |
| *Referring Expression Segmentation (RES)* | | | | | | | | | | | | |
| LoRA(r=16) | 28.0 | 223.4 | 12.5% | 78.1 | 78.9 | 76.7 | 69.1 | 72.4 | 62.8 | 69.8 | 70.1 | 72.2 |
| LoRA(r=54) | 40.6 | 236.0 | 17.2% | 78.4 | 79.6 | 77.2 | 69.1 | 72.9 | 62.6 | 69.3 | 69.4 | 72.3 |
| CoLA(r=16) | 40.5 | 236.0 | 17.2% | **79.3** | **80.3** | **77.5** | **70.6** | **74.6** | **64.6** | **71.3** | **71.4** | **73.7** |

*Table 2.* Comparison of CoLA and LoRA on audio-visual tasks with rank-matched (r=16) and parameter-matched configurations. CoLA consistently outperforms LoRA on both AVE and AVS tasks, validating the effectiveness of its cross-modal adaptation.

| Method | Param Update | Total Param | Update Ratio | Avg ↑ |
|---|---|---|---|---|
| *Audio-Visual Event Localization (AVE)* | | | | |
| LoRA(r=16) | 6.1 | 183.1 | 3.3% | 79.2 |
| LoRA(r=54) | 18.7 | 195.6 | 9.6% | 79.2 |
| CoLA(r=16) | 18.6 | 195.6 | 9.5% | **80.7** |
| *Audio-Visual Segmentation (AVS)* | | | | |
| LoRA(r=16) | 28.9 | 348.1 | 8.3% | 80.1 |
| LoRA(r=48) | 44.6 | 363.8 | 12.3% | 80.2 |
| CoLA(r=16) | 44.8 | 364.0 | 12.3% | **80.9** |

and fed into a trainable linear classifier. For AVS, we utilize SwinV2-L (Liu et al., 2022) as the vision backbone and SSLAM (Alex et al., 2025) as the audio backbone, adopting the segmentation decoder from (Zhou et al., 2022) and replacing their original backbone with our chosen backbone. All backbone encoders remain frozen during training, with only the downstream modules being trainable. We applied CoLA to all Q, K, V, and FFN components of both backbones. We use the rank of 16 for both intra- and inter-modal pathways in CoLA. For DINO-B configurations, CoLA is applied to all transformer layers. For Swin-L, we apply CoLA evenly distributed across layers to match the layer count of the SSLAM audio backbone, while the remaining layers use LoRA. For further implementation details and training settings for both tasks, refer to Appendix A.2.

## 4.2. Comparision with LoRA

We compare CoLA with two LoRA baselines: same rank (r=16) and increased rank to match CoLA's parameter count. The same-rank comparison is to demonstrate whether CoLA's cross-modal architecture is inherently superior with identical adaptation capacity. Since CoLA introduces ad-

ditional parameters through inter-modal fusion matrices and hypernetwork components, the same-parameter comparison ensures improvements are not simply due to having more parameters. The results are presented in Table 1 and 2. CoLA consistently outperforms LoRA in both comparisons across all tasks. For the same-rank comparison (r=16), CoLA achieves average improvements of 1.1% and 1.5% on vision-language tasks and 1.5% and 0.8% on audio-visual tasks. Even when LoRA uses increased rank to match CoLA's parameter count, CoLA maintains superior performance with average improvements of 1.6% and 1.4% on vision-language and 1.5% and 0.7% on audio-visual.

## 4.3. Comparison with previous work

To provide a comprehensive evaluation, we further compare CoLA's performance with existing PEFT methods, which are specifically designed for their respective multimodal downstream tasks in dual-encoder settings. While these methods employ task- or modality-specific architectural designs, CoLA achieves competitive performance through its dual low-rank pathway design, enabling effective cross-modal learning across different multimodal scenarios.

### 4.3.1. VISION-LANGUAGE TASKS

We compare the results of REC and RES with existing single-task PEFT methods and additionally include single-task and multi-task full-fine tuning (FT) approaches for comprehensive evaluation, presented in Table 3. Our result with CoLA establishes the first multi-task visual grounding using PEFT. For REC, CoLA achieves 83.4% average accuracy across RefCOCO datasets, outperforming FT baseline EEVG and other FT methods (VG-LAW, TransVG++, QR-Net, TransVG). Among PEFT methods, CoLA significantly outperforms SwimVG, HiVG, and MaPPER while using a 17.2% parameter update ratio. For RES, CoLA achieves 73.7% average mIoU, outperforming FT baseline EEVG and other FT methods (VG-LAW, CoupAlign, LAVT, CRIS).

*Table 3.* Results of different methods on Referring Expression Comprehension (REC) and Segmentation (RES) across RefCOCO, RefCOCO+, and RefCOCOg datasets. The Update Ratio column indicates the percentage of trainable parameters relative to total parameters in the whole model.

| Method | Update Ratio | RefCOCO | | | RefCOCO+ | | | RefCOCOg | | Avg ↑ |
|---|---|---|---|---|---|---|---|---|---|---|
| | | val | testA | testB | val | testA | testB | val | test | |
| *Referring Expression Comprehension (REC)* | | | | | | | | | | |
| TransVG (Deng et al., 2021) | 100% | 81.0 | 82.7 | 78.4 | 64.8 | 70.7 | 56.9 | 68.7 | 67.7 | 71.4 |
| TransVG++ (Deng et al., 2023) | 100% | 86.3 | 88.4 | 81.0 | 75.4 | 80.5 | 66.3 | 76.2 | 76.3 | 78.8 |
| QRNet (Ye et al., 2022) | 100% | 84.0 | 85.9 | 82.3 | 72.9 | 76.2 | 63.8 | 73.0 | 72.5 | 76.3 |
| VG-LAW (Su et al., 2023b) | 100% | 86.6 | 89.3 | 83.2 | 76.4 | 81.0 | 67.5 | 76.9 | 77.0 | 79.7 |
| EEVG (Chen et al., 2024) | 100% | 88.1 | 90.3 | 85.5 | 78.0 | 82.4 | 69.2 | 79.6 | 80.2 | 81.7 |
| HiVG (Xiao et al., 2024) | 20.1% | 87.3 | 89.9 | 83.3 | 78.1 | 83.8 | 68.1 | 78.3 | 78.8 | 80.9 |
| MaPPER (Liu et al., 2024) | 6.2% | 86.0 | 88.9 | 81.2 | 74.9 | 81.1 | 65.7 | 76.3 | 75.8 | 78.7 |
| SwimVG (Shi et al., 2025) | 2.04% | 88.3 | 90.4 | 84.9 | 77.9 | 83.2 | 69.95 | 80.1 | 79.7 | 81.8 |
| CoLA (Ours) | 17.2% | **89.4** | **91.0** | **86.9** | **79.6** | **84.7** | **71.9** | **81.7** | **81.8** | **83.4** |
| *Referring Expression Segmentation (RES)* | | | | | | | | | | |
| CRIS (Wang et al., 2022) | 100% | 70.5 | 73.2 | 66.1 | 62.3 | 68.1 | 53.7 | 59.9 | 60.4 | 64.3 |
| LAVT (Yang et al., 2022) | 100% | 72.7 | 75.8 | 68.8 | 62.1 | 68.4 | 55.1 | 61.2 | 62.1 | 65.8 |
| CoupAlign (Zhang et al., 2022) | 100% | 74.7 | 77.8 | 70.6 | 62.9 | 68.3 | 56.7 | 62.8 | 62.2 | 67.0 |
| VG-LAW (Su et al., 2023b) | 100% | 75.6 | 77.5 | 72.9 | 66.6 | 70.4 | 58.9 | 65.6 | 66.1 | 69.2 |
| EEVG (Chen et al., 2024) | 100% | 78.2 | 79.3 | 76.6 | 69.0 | 72.7 | 62.3 | 69.2 | 70.0 | 72.2 |
| ETRIS (Xu et al., 2023) | 17.4% | 70.5 | 73.5 | 66.6 | 60.1 | 66.9 | 50.2 | 59.8 | 59.9 | 63.4 |
| BarLeRIa (Wang et al., 2024b) | 17.8% | 72.4 | 75.9 | 68.3 | 65.0 | 70.8 | 56.9 | 63.4 | 63.8 | 67.1 |
| DETRIS (Huang et al., 2025) | 17.5% | 76.0 | 78.2 | 73.5 | 68.9 | 74.0 | 61.5 | 67.9 | 68.1 | 71.0 |
| CoLA (Ours) | 17.2% | **79.3** | **80.3** | **77.5** | **70.6** | **74.6** | **64.6** | **71.3** | **71.4** | **73.7** |

Among PEFT methods, CoLA substantially outperforms ETRIS, BarLeRIa, and DETRIS. ETRIS and BarLeRIa utilize multimodal pre-trained CLIP encoders, while DETRIS uses DINO for vision but retains CLIP's text encoder. Our approach instead uses completely separate unimodal pretrained models, highlighting the effectiveness of fully independent foundation models with cross-modal integration.

### 4.3.2. AUDIO-VISUAL TASKS

Both the results of AVE and AVS are presented in Table 4. For AVE, we compare our CoLA with existing PEFT methods that utilize ViT-based architectures. For a comprehensive comparison, we evaluate CoLA with additional architectures, including ViT-B-16 (pretrained on ImageNet) and DINOv2-L-14, and compare them against LAVisH's ViT-B-16 and STG-CMA's CLIP-L-14. With ViT-B-16 architectures, CoLA achieves 79.1% accuracy compared to LAVisH, and outperforms STG-CMA's CLIP-B-16. For a more appropriate comparison with CLIP, our DINOv2-B-14 achieves 80.7%, significantly outperforming STG-CMA's CLIP-B-16. When scaling to larger models, STG-CMA with CLIP-L-14 achieves 83.3% while our CoLA with DINOv2-L-14 reaches 81.1%. This gap arises from STG-CMA's specialized temporal and spatial adapters that benefit more from scaling. For AVS, we compare our CoLA to existing PEFT methods, LAVisH and STG-CMA, which use

shared Swin-L backbones, and DG-SCT, which uses separate encoders with Swin-L/HTS-AT. CoLA achieves 80.9% IoU, surpassing LAVisH and achieving comparable results with DG-SCT, while STG-CMA achieves the highest performance at 81.8%. Notably, DG-SCT employs specialized spatial-channel-temporal adapters for audio-visual modeling, while CoLA achieves comparable performance with a simpler design. Overall, CoLA achieves competitive performance across audio-visual tasks using separate unimodal foundation models. Despite its simpler cross-modal integration design, CoLA maintains competitive results.

## 5. Ablation Studies

### 5.1. Inter-modal and Intra-modal Pathway Design

We investigate sharing strategies for low-rank matrices between CoLA pathways, evaluating fully shared, partially shared, and fully non-shared configurations. The results are presented in Table 5. The fully shared configuration creates a single forward pathway where cross-modal features are integrated through the $\Phi$ matrix with standard LoRA. The two partially shared configurations establish distinct intra-modal and inter-modal pathways, while the fully non-shared configuration uses completely separate pathways (See Appendix C.1 for more details). All configurations with pathway separation outperform the fully shared baseline, with the fully

*Table 4.* Results of different methods on Audio-Visual Event Localization (AVE) and Audio-Visual Segmentation (AVS). Parameter Update and Total Parameters represent parameter counts measured in millions. The Update Ratio column indicates the percentage of trainable parameters relative to total parameters in the whole model.

| Method | Backbone Vision/Audio | Param Update | Total Parameters | Update Ratio | Metric Score ↑ |
|---|---|---|---|---|---|
| *Audio-Visual Event Localization (AVE)* | | | | | |
| LAVisH (Lin et al., 2023) | ViT-B-16 (Shared) | 4.7 | 107.2 | 4.4% | 75.3 |
| | ViT-L-14 (Shared) | 14.5 | 340.1 | 4.3% | 78.1 |
| STG-CMA (Wang et al., 2024a) | CLIP-B-16 (Shared) | 11.5 | 97.5 | 11.8% | 78.7 |
| | CLIP-L-14 (Shared) | 20.1 | 323.6 | 6.2% | **83.3** |
| | ViT-B-16/SSLAM | 18.6 | 211.6 | 8.8 | 79.1 |
| CoLA (Ours) | DINOv2-B-14/SSLAM | 18.6 | 195.6 | 9.5% | 80.7 |
| | DINOv2-L-14/SSLAM | 26.4 | 421.2 | 6.3% | 81.1 |
| *Audio-Visual Segmentation (AVS)* | | | | | |
| LAVisH (Lin et al., 2023) | Swin-L (Shared) | 37.2 | 266.4 | 14.0% | 80.1 |
| STG-CMA (Wang et al., 2024a) | Swin-L (Shared) | 38.6 | 233.6 | 16.5% | **81.8** |
| DG-SCT (Duan et al., 2023) | Swin-L/HTS-AT | 61.5 | 594.8 | 10.3% | 80.9 |
| CoLA (Ours) | Swin-L/SSLAM | 44.8 | 364.0 | 12.3% | 80.9 |

*Table 5.* Ablation study on sharing low-rank matrices between intra-modal and inter-modal pathways. LoRA Parameters represent parameter counts measured in millions. "Avg" is the average of validation and test performance on RefCOCOg.

| Shared B | Shared A | LoRA Parameters | REC Avg | RES Avg |
|---|---|---|---|---|
| ✓ | ✓ | 12.5 | 81.2 | 70.4 |
| ✗ | ✓ | 15.2 | 81.1 | 70.7 |
| ✓ | ✗ | 15.2 | 81.4 | 71.0 |
| ✗ | ✗ | 17.8 | **81.7** | **71.3** |

*Table 6.* Comparison of cross-modal feature propagation strategies on RefCOCOg validation and test sets. Progressive propagation outperforms both uniform and module-wise approaches, demonstrating the benefit of dynamically updating cross-modal information as it flows through the dual-encoder architecture.

| Method | REC | | RES | | Avg ↑ |
|---|---|---|---|---|---|
| | val | test | val | test | |
| Uniform | 81.3 | 81.2 | 70.9 | 70.9 | 76.1 |
| Module-wise | 81.0 | 81.5 | 70.7 | 71.1 | 76.1 |
| Progressive | **81.7** | **81.8** | **71.3** | **71.4** | **76.5** |

non-shared approach achieving the best results. This reveals that separating the low-rank matrices enables specialized projection mappings that capture different aspects of the input for modality-specific processing versus cross-modal fusion, thereby benefiting the cross-modal adaptation process.

### 5.2. Cross-modal Feature Propagation Strategy

We investigate different strategies for propagating cross-modal features to CoLA components throughout the dual-encoder architecture. We compare three propagation strategies for cross-modal features. These include Uniform (same cross-modal features across all components), Module-wise (identical features within each module type), and Progressive (features updated sequentially through component stages). For detailed diagrams and further implementation specifics of these strategies, refer to Appendix C.2. The results are presented in Table 6. The results show that progressive propagation achieves the best performance with an average score of 76.5%. Both uniform and module-wise strategies achieve identical average performance, though with slightly different distributions across tasks. The supe-

rior performance of progressive propagation demonstrates the effectiveness of continuously updating cross-modal information as it flows through the architecture. This approach enables each component to receive the most relevant and refined cross-modal features from previous stages, allowing for more sophisticated cross-modal integration.

### 5.3. Analysis of Cross-modal Influence

We investigate the influence of cross-modal through learned scaling factors $\lambda$ that control the contribution of inter-modal fusion in CoLA across different transformer layers and components. Figure 3 visualizes the scaling factors for vision-language and audio-visual tasks. For audio-visual, we present CoLA results in AVE. These scaling factors allow the cross-modal adaptation to selectively control where cross-modal information is most beneficial, enabling the model to learn to increase influence in layers that benefit from cross-modal interaction while reducing it in components where it may be unnecessary. In vision-language tasks, Q and K projections show higher scaling in earlier layers as CoLA enhances visual self-attention with language features, benefiting the model to identify relevant image re-

gions in visual grounding tasks, while other components demonstrate progressively increasing influence in deeper layers. In audio-visual, AVE demonstrates contrasting patterns, with lower early-layer cross-modal influence since the task requires semantic-level understanding, leading to progressively increasing scaling in deeper layers where semantic representations are formed and cross-modal fusion becomes most beneficial for the task.

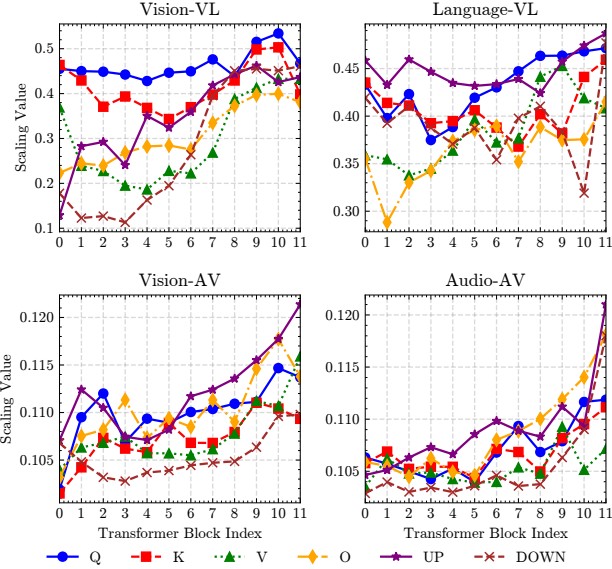

*Figure 3.* Visualization of learned scaling factors $\lambda$ across transformer layers for different components ($W_q$, $W_k$, $W_v$, $W_o$, $W_{up}$, $W_{down}$) in dual-encoder architectures. The plots show how cross-modal interaction strength varies by layer depth and component type for vision-language and audio-visual tasks, with higher $\lambda$ values indicating stronger cross-modal influence.

### 5.4. Rank and Parameter Optimisations in CoLA

The hypernetwork's up-projection $\mathbf{W}_{up} \in \mathbb{R}^{r^2 \times d_c / \gamma}$ scales with $r^2$, dominating cost as $r$ grows, and the two pathways are not constrained to share the same rank. We explore two design choices to mitigate the parameter scaling.

**Kronecker decomposition of $\Phi$.** Instead of producing $\Phi \in \mathbb{R}^{r \times r}$ directly, we factor it as $\Phi = \Phi_1 \otimes \Phi_2$ with $\Phi_1 \in \mathbb{R}^{m \times p}$ and $\Phi_2 \in \mathbb{R}^{n \times q}$, where $mn = pq = r$. The hypernetwork now outputs the entries of $\Phi_1$ and $\Phi_2$ rather than the full $r \times r$ matrix, so $\mathbf{W}_{up}$ produces $mn + pq$ values instead of $r^2$, an $8\times$ reduction at $r = 16$ and a $13\times$ reduction at $r = 32$, the reconstructed $\Phi$ still has shape $r \times r$. See Appendix C.4 for a detailed parameter breakdown.

**Asymmetric rank allocation.** The inter-modal pathway is the one that produces the $r^2$ expressive capacity, while the intra-modal pathway behaves as a standard LoRA branch. We therefore explore allocating a larger rank to the inter-

modal pathway and lowering the intra-modal rank.

We evaluate both strategies on AVE; results are reported in Table 7. Kronecker at the symmetric setting keeps the advantage over LoRA while shrinking $\mathbf{W}_{up}$ by $8\times$. Combining Kronecker with asymmetric ranks achieves the best result, surpassing the $r^2$ variant at a fraction of its $\mathbf{W}_{up}$ size. Therefore, allocating a larger rank to the inter-modal pathway makes better use of the parameter budget than splitting the rank evenly between both pathways.

*Table 7.* Strategies for reducing the parameter cost of CoLA's hypernetwork, evaluated on AVE. We compare LoRA baselines against CoLA with $\Phi$ as an $r^2$ matrix and as a Kronecker factorisation. $\mathbf{W}_{up}$ params reports the output dimension of the hypernetwork's up-projection.

| Method | $\Phi$ | Intra $r$ | Inter $r$ | $\mathbf{W}_{up}$ | Acc |
|---|---|---|---|---|---|
| LoRA | — | 16 | — | — | 79.2 |
| LoRA | — | 54 | — | — | 79.2 |
| CoLA | $r^2$ | 16 | 16 | 256 | 80.7 |
| CoLA | $mn + pq$ | 16 | 16 | 32 | 79.9 |
| CoLA | $mn + pq$ | 8 | 32 | 80 | **81.44** |

## 6. Conclusion

We propose CoLA, which extends the capability of low-rank adaptation with cross-modal integration through separate intra- and inter-modal pathways. CoLA addresses LoRA's limitation in lacking cross-modal interaction by enabling cross-modal awareness between unimodal encoders for multimodal tasks. Furthermore, we introduce progressive cross-modal propagation to facilitate continuous information exchange between dual encoders. We provide extensive experiments across vision-language and audio-visual tasks to validate CoLA's effectiveness over LoRA and show competitive performance against existing specialized PEFT methods. Additionally, CoLA enables the first multi-task visual grounding approach using PEFT. Lastly, we provide ablation studies that confirm that separate pathways and progressive propagation are crucial for optimal cross-modal adaptation. This work opens new directions for cross-modal LoRA adaptation, demonstrating effective integration of cross-modal information within the low-rank adaptation paradigm. Future work could explore using other LoRA variants in CoLA or integrating CoLA into LLMs to enable multimodal capabilities, transforming them into Multimodal LLMs.

**Limitations:** During inference, the intra-modal pathway can be merged with pre-trained weights following standard LoRA practices, eliminating computational overhead. In contrast, the inter-modal pathway cannot be merged, as it depends on dynamic cross-modal features in the dual-encoder architecture. Please see Appendix D for additional details on these limitations.

# Impact Statement

This paper presents a method for efficient cross-modal adaptation that enables foundation models from different modalities to work together effectively on multimodal tasks. The primary societal benefit is democratizing access to multimodal AI by reducing computational requirements while enhancing downstream task performance through cross-modal information exchange. Our approach allows any combination of modalities (vision, language, audio) to be adapted efficiently for specific applications, making advanced multimodal AI more accessible to researchers and organizations with limited resources.

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

# A. Experimental Setting

## A.1. Vision-Language Task

For training setup, we freeze both vision and text backbones and train only the multi-task decoder for REC and RES along with PEFT modules, using separate learning rates for each module. Note that in CoLA, we use the same rank for low-rank matrices in both intra-modal and inter-modal pathways, and both CoLA settings are applied identically to both vision and text backbones. These settings are applied consistently across the training of all RefCOCO datasets. The hyperparameter settings are detailed in Table 8.

*Table 8.* Training settings for CoLA on Vision-Language tasks.

| Hyperparameters | |
| --- | --- |
| Rank $r$ | 16 |
| Scaling $\alpha$ | 8 |
| Scaling $\lambda$ | 0.5 |
| Reduction $\gamma$ | 16 |
| Optimizer | AdamW |
| Weight Decay | $1 \times 10^{-4}$ |
| LR Adapter | $1 \times 10^{-4}$ |
| LR Decoder | $2.5 \times 10^{-5}$ |
| LR Scheduler | Polynomial |
| Poly Power | 0.9 |
| Epochs | 150 |
| Batch Size | 80 |
| Image Size | 448 |

## A.2. Audio-Visual Tasks

### A.2.1. AUDIO-VISUAL EVENT LOCALIZATION (AVE)

For AVE, we freeze both vision and audio backbone encoders and train only the linear classifier along with CoLA components, using separate learning rates for different modules. CoLA settings are applied identically to vision encoders (ViT-B-16, DINO-B-14, DINO-L-14) and SSLAM audio encoder. For ViT-B-16 and DINO-B-14 with 12 layers, all layers are paired with SSLAM's 12 layers for cross-modal fusion. For DINO-L-14 with 24 layers, CoLA is applied to even layers matching SSLAM layers, while LoRA is applied to odd layers. The training hyperparameters for AVE are detailed in Table 9.

*Table 9.* Training settings for Audio-Visual Event Localization

| Hyperparameters | Value |
| --- | --- |
| Rank $r$ | 16 |
| Scaling $\alpha$ | 8 |
| Scaling $\lambda$ | 0.1 |
| Optimizer | Adam |
| LR Adapter | $5 \times 10^{-6}$ |
| LR MLP | $4 \times 10^{-6}$ |
| Epochs | 50 |
| Batch Size | 2 |

### A.2.2. AUDIO-VISUAL SEGMENTATION (AVS)

For AVS, we freeze both vision and audio backbone encoders and train only the segmentation decoder along with CoLA components. Swin-L consists of 4 stages with a total of 24 layers. CoLA is applied to even layers matching SSLAM layers, while LoRA is applied to odd layers. CoLA settings are applied identically to Swin-L vision encoder and SSLAM audio encoder, except for the reduction factor $\gamma$. The training hyperparameters for AVS are detailed in Table 10. For the reduction factor $\gamma$, each of the 4 Swin-L stages has different feature dimensions, so $\gamma$ is adjusted across these stages, starting from

*Table 10.* Training settings for Audio-Visual Segmentation

| Hyperparameters | Value |
|---|---|
| Rank $r$ | 16 |
| Scaling $\alpha$ | 8 |
| Scaling $\lambda$ | 0.1 |
| Optimizer | Adam |
| LR | $2 \times 10^{-4}$ |
| Epochs | 15 |
| Batch Size | 8 |

$\gamma = 2$ for the first stage and progressing as [2,4,8,16] across the 4 Swin-L stages to CoLA in SSLAM, while SSLAM to CoLA in Swin uses a fixed reduction factor of 16.

## B. Dataset Details

### B.1. Vision-Language Dataset

#### B.1.1. REFCOCO

contains 19,994 images with 142,210 referring expressions describing 50,000 objects. The dataset is split into four subsets with training, validation, testA, and testB samples. Each image contains an average of at least two objects, with referring expressions averaging 3.6 words in length. We trained the model on the training set and reported the result on the validation and test sets.

#### B.1.2. REFCOCO+

contains 19,992 images with 141,564 referring expressions linked to 49,856 objects, with expressions excluding absolute-location words. The dataset is split into four subsets with training, validation, testA, and testB samples. We trained the model on the training set and reported the result on the validation and test sets.

#### B.1.3. REFCOCOG

contains 25,799 images with 141,564 referring expressions associated with 49,856 objects, featuring longer and more complex language expressions. We utilized the UMD-split for RefCOCOg, which partitions the data into training, validation, and test sets. We trained the model on the training set and reported the result on the validation and test sets.

### B.2. Audio-Visual Dataset

#### B.2.1. AUDIO-VISUAL EVENT LOCALIZATION (AVE) DATASET

consists of 4,143 videos, each with a 10-second duration and annotations marking the temporal boundaries of audio-visual events, where each second is labeled across 28 event categories. The dataset is split into training, validation, and test sets. We trained the model on the training set and reported the result on the test set.

#### B.2.2. AUDIO-VIUAL SEGMENTATION (AVS) DATASET (AVSBENCH-S4)

consisting of 4,932 videos with manual pixel-level segmentation mask annotations of audible objects of over 23 categories. The dataset is split into training, validation, and test sets. We trained the model on the training set and reported the result on the test set.

## C. Ablations Study

### C.1. Inter-modal and Intra-modal Pathway Design

The illustration of different pathway designs is shown in Figure 4. When either B or A matrices (or both) are non-shared, the experiment employs two distinct forward passes: one for intra-modal adaptation and another for inter-modal fusion.

When both B and A matrices are shared between pathways, this reduces to a single unified forward pass that handles both processing. In this shared configuration, we do not use the learnable scaling parameter $\lambda$ and instead use a static LoRA scaling factor $\alpha$.

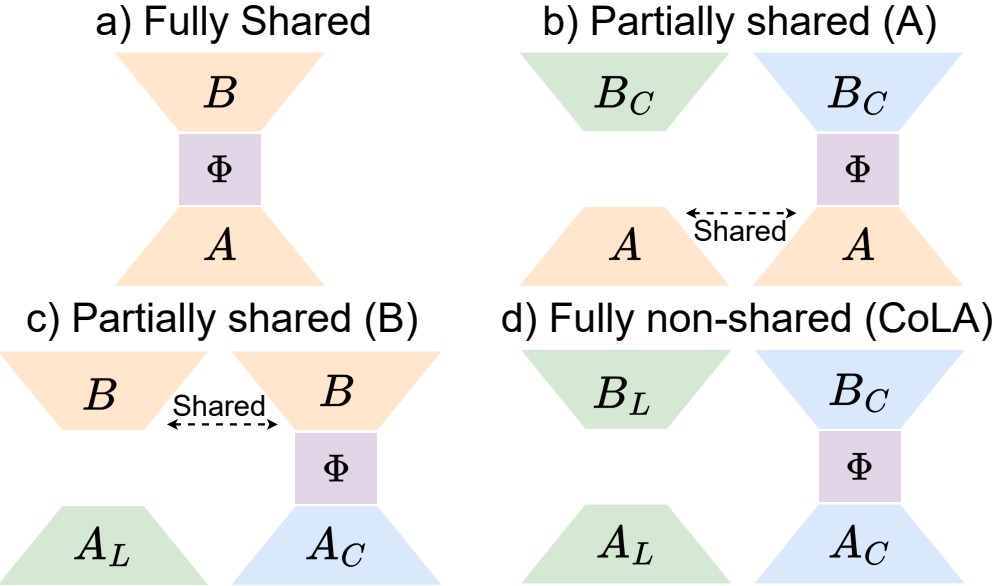

*Figure 4.* Illustration of different sharing strategies for CoLA low-rank matrices between pathways: (a) Fully shared, (b) Partially shared A, (c) Partially shared B, (d) Fully non-shared.

## C.2. Cross-modal Feature Propagation Strategy

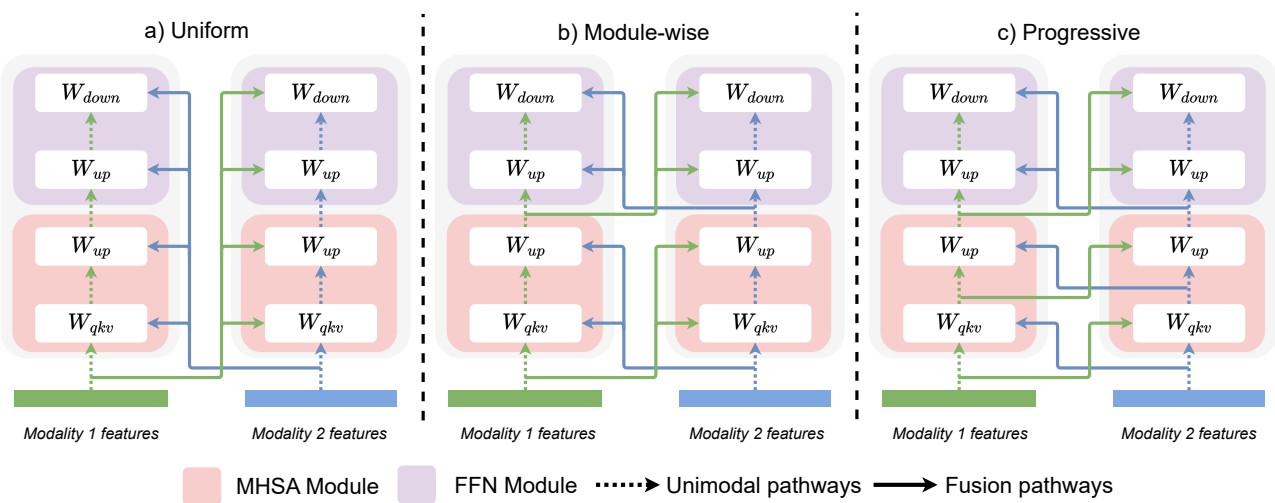

*Figure 5.* Comparison of cross-modal propagation strategies: (a) uniform, (b) module-wise, and (c) progressive designs

The uniform and module-wise propagation strategies are defined as illustrated in Figure 5. In the uniform design, the same cross-modal features from the paired encoder are shared across all CoLA components within that layer. In the module-wise design, the cross-modal input is shared across CoLA in MHSA, and the cross-modal output from MHSA is exchanged between dual encoders to serve as cross-modal input for CoLA in the FFN module.

## C.3. Unidirectional vs Bidirectional Cross-Modal Adaptation

CoLA enables symmetric bidirectional cross-modal interaction, where both modalities simultaneously adapt based on features from the paired modality. To validate the importance of this bidirectional design, we compare bidirectional CoLA against unidirectional variants on the AVE task. In the unidirectional variants, only one modality is adapted using cross-modal features from the other, while the paired modality is adapted using standard LoRA. The results are presented in Table 11.

*Table 11.* Comparison of unidirectional and bidirectional cross-modal adaptation on the AVE task.

| Method | Direction | AVE |
|--------|-----------|-----|
| LoRA | – | 79.2 |
| CoLA | Vision to Audio only | 79.6 |
| CoLA | Audio to Vision only | 80.2 |
| CoLA | Bidirectional | **80.7** |

Bidirectional cross-modal adaptation consistently outperforms both unidirectional variants, demonstrating that symmetric cross-modal interaction between both modalities is essential to CoLA's performance gains. Both unidirectional variants still outperform LoRA, confirming that even one-directional cross-modal adaptation provides benefits over modality-isolated adaptation. However, the bidirectional design enables both modalities to mutually inform each other's adaptation, leading to the best overall performance.

## C.4. Parameter Cost of Kronecker Decomposition

Section 5.4 introduces the Kronecker decomposition and asymmetric rank allocation as strategies to control the $\mathcal{O}(r^2)$ scaling of the hypernetwork's up-projection $\mathbf{W}_{up}$. Here, we provide a detailed parameter breakdown of the hypernetwork across different configurations, including the contribution of the reduction factor $\gamma$ in the bottleneck. The full breakdown is reported in Table 12.

*Table 12.* Detailed parameter cost breakdown of the CoLA hypernetwork. $\mathbf{W}_{down}$ compresses cross-modal features from $d_c$ to $d_c/\gamma$, and $\mathbf{W}_{up}$ projects to the $\Phi$ representation. The configuration without $\gamma$ uses a direct projection from $d_c$ to $r^2$. Parameters are reported assuming $d_c = 768$ and $\gamma = 16$.

| Method | $r$ | $\mathbf{W}_{down}$ Params | $\mathbf{W}_{up}$ Params | Total Params |
|--------|-----|----------------------------|--------------------------|--------------|
| MLP (no $\gamma$) | 16 | – | 196,608 | 196,608 |
| MLP + $\gamma$ | 16 | 36,864 | 12,288 | 49,152 |
| Kronecker + $\gamma$ | 16 | 36,864 | 1,536 | 38,400 |
| MLP + $\gamma$ | 32 | 36,864 | 49,152 | 86,016 |
| Kronecker + $\gamma$ | 32 | 36,864 | 3,840 | 40,704 |

The reduction factor $\gamma$ alone compresses the hypernetwork through the bottleneck before $\Phi$ is reconstructed, reducing total parameters by approximately 4x at $r = 16$ (from 196,608 to 49,152). Combining $\gamma$ with the Kronecker decomposition introduced in Section 5.4 further compresses $\mathbf{W}_{up}$ from 12,288 to 1,536 parameters at $r = 16$. At $r = 32$, the gap widens: Kronecker keeps $\mathbf{W}_{up}$ at 3,840 parameters while a standard MLP would require 49,152. These results show that the $\mathcal{O}(r^2)$ cost of the hypernetwork can be controlled in practice, supporting CoLA's applicability at higher ranks.

# D. Limitations

CoLA introduces minor computational overhead compared to standard LoRA due to the inter-modal pathway's reliance on dynamic cross-modal features. Unlike the intra-modal pathway, which can be merged into pre-trained weights following standard LoRA practices, the inter-modal pathway must compute cross-modal interactions at runtime. In the inference comparison shown in Figure 6, the intra-modal pathway weights of CoLA are fully merged into the pre-trained weights, following the same standard LoRA practice applied to the LoRA baseline. The remaining overhead in CoLA therefore comes solely from the inter-modal pathway. As shown in Figure 6, this results in modest increases in memory usage, training time, and inference latency. However, these overhead costs are reasonable trade-offs for the performance benefits CoLA provides, and the improved model quality justifies the marginal computational expense.

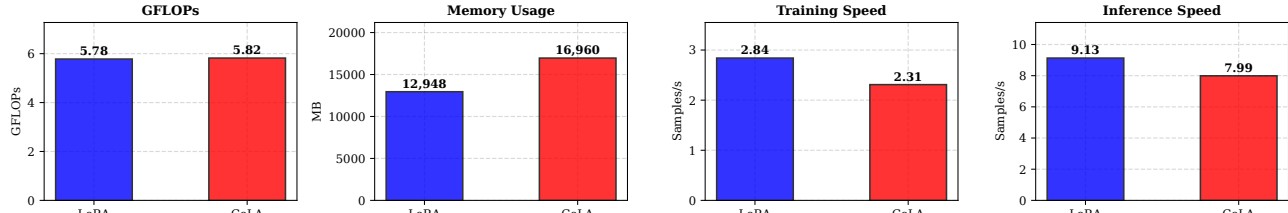

*Figure 6.* Visualization of computational and memory trade-offs experiment between LoRA and CoLA on the AVE task. While CoLA achieves comparable computational efficiency (GFLOPs), the inter-modal pathway cannot be merged into pre-trained weights, which introduces modest runtime overhead at inference. CoLA results in increases in GPU memory (MB), training time (samples/s), and inference latency (samples/s) compared to LoRA, costs due to the dynamic cross-modal feature computation required during both training and inference.

