# OpenReview forum: "CoLA: Cross-Modal Low-rank Adaptation for Multimodal Downstream Tasks"
_ICML.cc/2026/Conference — ICML 2026 regular_

### Official Review · Reviewer_u5qe · 2026-02-27

**Soundness:** 2
**Presentation:** 2
**Significance:** 2
**Originality:** 2
**Overall Recommendation:** 2
**Confidence:** 5

**Summary:**

The authors identify a key limitation of applying Low-Rank Adaptation (LoRA) to multimodal dual-encoder architectures: LoRA operates in isolation within each modality, lacking cross-modal awareness and thus failing to leverage complementary information between different modalities for downstream tasks. To address this gap, the authors propose Cross-Modal Low-rank Adaptation (CoLA), a novel Parameter-Efficient Fine-Tuning (PEFT) framework that extends LoRA with a dedicated inter-modal fusion pathway alongside the standard intra-modal adaptation pathway. This dual-path design enables bidirectional cross-modal interaction at linear components of transformer modules while maintaining a clear separation between modality-specific and cross-modal computations. A central topic explored by this study is the integration of cross-modal awareness into low-rank adaptation paradigms, with additional investigations into cross-modal feature propagation strategies and the influence of cross-modal scaling factors across transformer layers. The experiments show the effectiveness of the method.

**Compliance With Llm Reviewing Policy:**

Affirmed.

**Final Justification:**

Thank you to the authors for the detailed clarification and additional experiments. While the rebuttal presents several reasonable optimization strategies that modestly alleviate the scaling issue, my concern is only partially addressed: the method appears to require multiple design modifications to control overhead, yet even with these optimizations it still does not reach the parameter efficiency of a standard LoRA-style adaptation. As a result, I remain unconvinced that the scalability concern has been fully resolved.

**Key Questions For Authors:**

1. Could the authors add an in-depth theoretical analysis of CoLA, including mathematical derivation and theoretical bounds of the cross-modal fusion pathway with the transformation matrix $\Phi^m$.
2. As CoLA’s parameter count scales with $r^2$ and surges with increasing rank $r$, have you explored optimization strategies (e.g., low-rank decomposition of $\Phi^m$, sparse design) to alleviate parameter explosion while maintaining fusion performance?
3. Fixed scaling factor and reduction factor are used in experiments. Could you explain their selection basis and whether task/modality-specific optimal ranges exist? Do these hyperparameters interact synergistically/antagonistically with rank $r$ on performance and efficiency?

**Limitations:**

yes

**Strengths And Weaknesses:**

Strengths:
1. The paper tries to address a practical limitation of LoRA in multimodal settings, with a clear problem formulation that fills a critical gap in efficient multimodal model adaptation.
2. The paper is easy to follow.

Weaknesses:
1. The number of parameters occupied by the CoLA method is proportional to $r^2$ for cross-modal fusion; as the low-rank matrix rank r increases to enhance model performance, the model's trainable parameters will grow rapidly, which may compromise the parameter efficiency advantage of CoLA in practical applications with higher rank requirements.
2. No efficiency comparisons with baselines are provided, including runtime, GPU memory and parameter count.
3. The paper mainly compares CoLA with the original LoRA, failing to include advanced LoRA derivatives such as QLoRA, LoRA+ and AdaLoRA. The comparative baselines also lack multimodal PEFT methods, such as MAPLE [1].
[1] Khattak M U, Rasheed H, Maaz M, et al. Maple: Multi-modal prompt learning[C]//Proceedings of the IEEE/CVF conference on computer vision and pattern recognition. 2023: 19113-19122.

---

> ### Author Rebuttal · Authors · 2026-03-27
>
> Dear Reviewer u5qe, Thank you for your feedback.
>
> **W1 & Q2:**
> - **Parameter scaling** The parameter scaling concern primarily comes from the hypernetwork generating the $r^2$ dimensional $\Phi$ matrix. In practice, CoLA achieves strong performance at r=16, where the hypernetwork overhead remains modest relative to the overall model size.
> - **Optimization strategies:** For higher rank scenarios, the MLP-based $\Phi$ generation can be replaced with a Kronecker product decomposition: instead of outputting $r^2$ values directly, we can generate two smaller matrices $\mathbf{P} \in \mathbb{R}^{m \times n}$ and $\mathbf{Q} \in \mathbb{R}^{p \times q}$ where $mp = nq = r$, and compute  $\Phi = \mathbf{P} \otimes \mathbf{Q} \in \mathbb{R}^{mp \times nq}$. This reduces hypernetwork output from $r^2$ to $mn + pq$ parameters (e.g., 256 to 32 for $r=16$, $m=n=p=q=4$).
>
> **W2:Efficiency comparisons** Parameter counts are reported in Tables 1, 2, and 4. Detailed efficiency comparisons (GFLOPs, memory, training/inference speed) are in Figure 6 (Appendix D). CoLA's GFLOPs are nearly identical to LoRA (5.78 vs 5.82), with modest increases in memory (12,948 vs 16,960 MB) and speed (2.84 vs 2.31 samples/s training, 9.13 vs 7.99 samples/s inference).
>
> **W3: Missing baselines** We thank the reviewer for this suggestion regarding broader baseline comparisons. We address both concerns below.
> - **On LoRA variants:** We chose standard LoRA as our primary baseline because CoLA is **designed as an extension of LoRA**, directly isolating the contribution of our cross-modal mechanism. Other LoRA variants represent **orthogonal improvements** to intra-modal adaptation, and CoLA's inter-modal pathway is **independent of the intra-modal LoRA variant**  making it compatible with any LoRA-style method.
> - **MAPLE:** MaPLE was not included as it targets different task domains than REC, RES, AVE, and AVS. We provide additional comparison on AVE using the same backbone and training settings:
>
> | Method | Accuracy |
> |--------|-------------|
> | MaPLE  | 78.93       |
> | CoLA   | **80.7**    |
>
> As shown in the table above, CoLA outperforms both multimodal PEFT methods on AVE, demonstrating the effectiveness of its cross-modal adaptation design over existing PEFT approaches for dual-encoder architectures.
>
>
> **Q1: Theoretical analysis of CoLA** We provide formal analysis of the cross-modal fusion pathway with $\Phi^m$ below:
> - **Theorem 1:** $\text{rank}(\Delta W_C^m) \leq \min(\text{rank}(B_C^m), \text{rank}(\Phi^m), \text{rank}(A_C^m), r)$. When $\Phi^m$ is full rank, it rotates and scales the low-rank subspace without introducing additional rank constraints. When rank-deficient, it acts as an implicit gating mechanism, selectively reducing the dimensionality of cross-modal interaction.
> - **Theorem 2:** Each individual update has rank $\leq r$ (Theorem 1), but different inputs produce different $\Phi^{(k)}$, rotating the internal mapping within the rank-$r$ subspace spanned by $B_C$ and $A_C$ per input. Vectorizing into weight space: $\text{vec}(\Delta W_C^{(k)}) = \lambda (A_C^\top \otimes B_C) \text{vec}(\Phi^{(k)})$, the linear map $(A_C^\top \otimes B_C)$ has rank $\leq \text{rank}(A_C) \cdot \text{rank}(B_C) \leq r^2$. So across $K$ inputs, CoLA spans up to $\min(K, r^2, d_{out} \cdot d_{in})$ dimensions in weight space. Each update remains rank-$r$ in activation space, but different $\Phi^{(k)}$ collectively span up to $r^2$ dimensional subspace of effective weight matrices compared to LoRA's single fixed $\Delta W$, which spans only $1$ direction in weight space and at most $r$ directions in activation space.
> - **Proposition 1 (Gradient Decoupling):** Given $\delta = \frac{\partial \mathcal{L}}{\partial h_m}$, the parameter gradients are:
>
> $$\frac{\partial \mathcal{L}}{\partial B_L^m} = \frac{\alpha}{r} \delta (A_L^m x_m)^\top \qquad \frac{\partial \mathcal{L}}{\partial B_C^m} = \lambda \delta (\Phi^m A_C^m x_m)^\top$$
> Intra-modal gradients depend only on $\{B_L^m, A_L^m, \delta\}$; inter-modal on $\{B_C^m, A_C^m, \Phi^m,  \lambda, \delta\}$. The pathways are structurally decoupled at the parameter level through disjoint computational branches,mitigating interference between pathways. This supports Table 5's finding that non-shared pathways perform best.
>
> **Q3: Hyperparameter selection** The scaling factor $\lambda$ is **learnable**, initialized at 0.5 (vision-language) and 0.1 (audio-visual), and optimized during training (Figure 3). Figure 3 shows $\lambda$ converges to task- and modality-specific values, suggesting optimal ranges vary across tasks and modalities. The reduction factor $\gamma$ is architecture-dependent: $\gamma=16$ for ViT-based encoders (constant feature dimension); adjusted per stage for Swin-based encoders (varying feature dimensions), as detailed in Appendix A.2.2. Both intra- and inter-modal pathways use $r=16$, though their ranks can be independently configured for further optimization.

---

> > ### Author Rebuttal · Reviewer_u5qe · 2026-04-01
> >
> > Setting the maximum rank r to 16 will severely limit the applicability of the model. The $r^2$-level parameter complexity deviates from the fundamental motivation of PEFT (Parameter-Efficient Fine-Tuning) and makes it difficult to apply in practice. Moreover, the authors’ optimization strategy lacks sufficient experimental support. I’m sorry to say that the rebuttal does not provide enough evidence to justify raising my score.

---

> > > ### Author Response · Authors · 2026-04-05
> > >
> > > Dear Reviewer u5qe,
> > >
> > > We thank the reviewer for the continued engagement and thoughtful follow-up. We provide the following clarification to address the $r^2$ scaling and experimental evidence of optimization strategy concern below.
> > >
> > > **$r^2$ Scaling concern**
> > >
> > > To achieve the $r^2$ expressivity established in Theorems 1 and 2 in our previous response, CoLA generates a full $r \times r$ matrix $\Phi$ via the MLP hypernetwork, which introduces $r^2$ parameter scaling. This scaling is already controlled and can be further optimized by CoLA's reduction factor $\gamma$ (in Equation 6), which introduces a bottleneck in the hypernetwork where the cross-modal feature $d_c$ is first compressed to $d_c/\gamma$ via $W_{down} \in \mathbb{R}^{d_c/\gamma \times d_c}$, before projecting to $r^2$ via $W_{up} \in \mathbb{R}^{r^2 \times d_c/\gamma}$. Notably, $\gamma$ can be further tuned per architecture and stage, as detailed in Appendix A.2.2, for Swin-based encoders with varying feature dimensions across stages, $\gamma$ is adjusted per stage, demonstrating that the hypernetwork cost can be flexibly controlled according to architectural requirements. To illustrate the impact of $\gamma$, the following table compares the hypernetwork parameter cost with and without $\gamma$, and further shows how Kronecker decomposition, mentioned in our previous response as an optimization strategy, provides additional reduction at $d_c=768$:
> > >
> > > | Method | $r$ | $W_{down}$ Params | $W_{up}$ Params | Total Params |
> > > |--------|-----|--------------------------|--------------------------|-------|
> > > | MLP (no $\gamma$) | 16 | - | 196,608 | 196,608 |
> > > | MLP + $\gamma=16$ | 16 | 36,864 | 12,288 | 49,152 |
> > > | Kronecker + $\gamma=16$ | 16 | 36,864 | 1,536 | 38,400 |
> > > | MLP + $\gamma=16$ | 32 | 36,864 | 49,152 | 86,016 |
> > > | Kronecker + $\gamma=16$ | 32 | 36,864 | 3,840 | 40,704 |
> > >
> > >
> > > **Optimization strategy with experimental validation**
> > >
> > > For higher rank scenarios where $r^2$ scaling becomes more of a concern, Kronecker decomposition can replace the $W_{up}$ projection, reducing its output from $r^2$ to $mn+pq$ where $mp=nq=r$, while $W_{down}$ remains identical. At $r=16$, $\gamma=16$, $d_c=768$ with $m=n=p=q=4$, Kronecker reduces $W_{up}$ from 12,288 to 1,536, a **8x reduction**. Crucially, at $r=32$ with $m=n=4$, $p=q=8$, Kronecker keeps $W_{up}$ at 3,840 while MLP would require 49,152, a **13x reduction**. This directly prevents the $r^2$ parameter scaling in $W_{up}$, making higher rank adaptation practically feasible without compromising PEFT efficiency. We validate this with additional experiments comparing MLP and Kronecker at symmetric ranks of 16 in the table below:
> > >
> > > | Method | $\Phi$ structure | Intra $r$ | Inter $r$ | $W_{up}$ output | AVE |
> > > |--------|-----------------|-----------|-----------|--------------------------|-----|
> > > | LoRA | - | 16 | - | - | 79.2 |
> > > | LoRA | - | 54 | - | - | 79.2 |
> > > | CoLA MLP | $r^2$ | 16 | 16 | 256 | 80.7 |
> > > | CoLA Kronecker | $mn+pq$ | 16 | 16 | 32 | 80.0 |
> > >
> > > Kronecker decomposition at the same rank already outperforms both LoRA baselines while reducing $W_{up}$ by **8x**. As noted in the paper, we use the same rank for both intra- and inter-modal pathways for simplicity and fairness. However, to further demonstrate CoLA's applicability at higher rank requirements as raised by the reviewer, we explore asymmetric rank allocation to further optimize the number of parameters, since CoLA's intra- and inter-modal ranks can be configured independently, the intra-modal rank can be compressed when the unimodal backbone requires less rank budget, dedicating more capacity to cross-modal expressivity:
> > >
> > >
> > > | Method | $\Phi$ structure | Intra $r$ | Inter $r$ | $W_{up}$ output | AVE |
> > > |--------|-----------------|-----------|-----------|--------------------------|-----|
> > > | LoRA | - | 16 | - | - | 79.2 |
> > > | CoLA MLP | $r^2$ | 16 | 16 | 256 | 80.7 |
> > > | CoLA Kronecker | $mn+pq$ | 16 | 16 | 32 | 80.0 |
> > > | CoLA Kronecker | $mn+pq$ | 8 | 32 | 80 | **81.4** |
> > >
> > > Kronecker with asymmetric ranks (Intra $r$=8, Inter $r$=32) achieves **81.4%**, outperforming baseline CoLA MLP (80.7%) while using only 80 $W_{up}$ parameters versus 256. At $r$=32, standard MLP would require 49,152 parameters while Kronecker uses 80, a **13x reduction** that prevents $r^2$ scaling. This demonstrates that the inter-modal pathway benefits from higher rank capacity while the intra-modal rank can be compressed. Symmetric $r$=16 already achieves strong performance (80.7%), and these optimizations enable higher ranks when needed.  We will include comprehensive analysis of Kronecker decomposition and asymmetric rank allocation in the camera-ready version.
> > >
> > > We hope these clarifications and experimental results sufficiently address the reviewer's concerns on $r^2$ scaling and the applicability of CoLA at higher ranks.

---

### Official Review · Reviewer_gkqp · 2026-03-04

**Soundness:** 3
**Presentation:** 3
**Significance:** 3
**Originality:** 2
**Overall Recommendation:** 4
**Confidence:** 3

**Summary:**

The authors propose a new cross-modal LoRA design for several downstream tasks. They enable the inter-modality pathway during training. They also demonstrate the effectiveness of this method in various datasets.

**Compliance With Llm Reviewing Policy:**

Affirmed.

**Final Justification:**

Please refer to ack.

**Key Questions For Authors:**

Please see the weakness.

**Limitations:**

N/A.

**Strengths And Weaknesses:**

Strength:

1-The research direction is reasonable and promising.

2-The motivation is clear.

3-The writing is clear and easy to follow.

4-Extensive experiments along with deep analysis are provided.

Weakness:

1-The method is not novel enough. The intermediate layer interaction between two paths is a widely explored design.

2-Lack a deep analysis behind the design. The experimental performances are good, but it would be better if the authors explain the reason from a theoretical perspective.

3-The authors should apply COLA to different backbones (encoders) to verify its roubustness.

---

> ### Author Rebuttal · Authors · 2026-03-27
>
> Dear Reviewer gkqp,
>
> Thank you for your detailed review and constructive feedback. We address each of your concerns below.
>
> **W1:Novelty** While intermediate layer interaction has been explored, prior methods are typically **unidirectional** (e.g., SwimVG: text to vision only), **task-specific** (e.g., SwimVG: REC only), and **modality-specific** (e.g., STG-CMA: audio-visual only), operating at the module level with task-specific designs. CoLA addresses these limitations:
>
> - **Weight-level adaptation:** CoLA enables cross-modal interaction at the weight level, directly adapting individual linear components through low-rank decomposition with cross-modal conditioning. This allows cross-modal information to be integrated at a finer granularity than module-level approaches, as formulated in Equation 4 and Algorithm 1 of our paper.
>
> - **Task, modality, and architecture generality:** Existing methods have only been demonstrated on a single task, a single modality pair, and a single backbone configuration. CoLA is **task-, modality-, and architecture-agnostic**  we demonstrate this across vision-language (REC, RES) and audio-visual (AVE, AVS) tasks with diverse backbones. We further demonstrate its backbones robustness in W3.
>
> To our knowledge, **CoLA is the first method** to extend LoRA with input-conditioned cross-modal adaptation specifically designed for dual-encoder architectures.
>
>
> **W2: Deeper analysis behind the design**
> We provide the following theoretical analysis on CoLA's rank, expressivity, and gradient behaviour:
> - **Theorem 1:** $\text{rank}(\Delta W_C^m) \leq \min(\text{rank}(B_C^m), \text{rank}(\Phi^m), \text{rank}(A_C^m), r)$. When $\Phi^m$ is full rank, it rotates and scales the low-rank subspace without introducing additional rank constraints. When rank-deficient, it acts as an implicit gating mechanism, selectively reducing the dimensionality of cross-modal interaction.
>
> - **Theorem 2:** Each individual update has rank $\leq r$ (Theorem 1), but different inputs produce different $\Phi^{(k)}$, rotating the internal mapping within the rank-$r$ subspace spanned by $B_C$ and $A_C$ per input. Vectorizing into weight space: $\text{vec}(\Delta W_C^{(k)}) = \lambda (A_C^\top \otimes B_C) \text{vec}(\Phi^{(k)})$, the linear map $(A_C^\top \otimes B_C)$ has rank $\leq \text{rank}(A_C) \cdot \text{rank}(B_C) \leq r^2$. So across $K$ inputs, CoLA spans up to $\min(K, r^2, d_{out} \cdot d_{in})$ dimensions in weight space. Each update remains rank-$r$ in activation space, but different $\Phi^{(k)}$ collectively span up to $r^2$ dimensional subspace of effective weight matrices compared to LoRA's single fixed $\Delta W$, which spans only $1$ direction in weight space and at most $r$ directions in activation space.
>
> - **Proposition 1 (Gradient Decoupling):** Given $\delta = \frac{\partial \mathcal{L}}{\partial h_m}$, the parameter gradients are:
> $$\frac{\partial \mathcal{L}}{\partial B_L^m} = \frac{\alpha}{r} \delta (A_L^m x_m)^\top \qquad \frac{\partial \mathcal{L}}{\partial B_C^m} = \lambda \delta (\Phi^m A_C^m x_m)^\top$$
> Intra-modal gradients depend only on $\{B_L^m, A_L^m, \delta\}$; inter-modal on $\{B_C^m, A_C^m, \Phi^m,  \lambda, \delta\}$. The pathways are structurally decoupled at the parameter level through disjoint computational branches,mitigating interference between pathways. This supports Table 5's finding that non-shared pathways perform best.
>
> **W3: Robustness across different backbones**
>
> Our paper already evaluates on diverse backbones: ViTDet+BERT (REC, RES), SwinV2-L+SSLAM (AVS), DINOv2-B/L+SSLAM (AVE), with CoLA outperforming LoRA across all (Tables 1, 2, 4). To further strengthen this evidence, we provide additional experiments on the AVE task with EVA-L [1] and DINOv3-B [2] as the vision backbone.
>
> | Vision Backbone | PEFT Method | AVE Performance |
> |-----------------|-------------|-----------------|
> | EVA-L           | LoRA        | 79.15           |
> | EVA-L           | CoLA        | **81.04**       |
> | DINOv3-B        | LoRA        | 79.90           |
> | DINOv3-B        | CoLA        | **81.66**       |
>
> CoLA consistently outperforms LoRA across both backbone architectures, achieving **+1.89%** with EVA-L and **+1.76%** with DINOv3-B. These additional results, combined with the backbone diversity already present in our paper, confirm that CoLA's improvements are **robust across different backbone architectures, modalities, and tasks**, and not dependent on a specific encoder choice.
>
> [1] Fang, Y., Wang, W., Xie, B., Sun, Q., Wu, L., Wang, X., Huang, T., Wang, X. and Cao, Y., 2023. Eva: Exploring the limits of masked visual representation learning at scale. In Proceedings of the IEEE/CVF conference on computer vision and pattern recognition (pp. 19358-19369).
>
> [2] Siméoni, O., Vo, H.V., Seitzer, M., Baldassarre, F., Oquab, M., Jose, C., Khalidov, V., Szafraniec, M., Yi, S., Ramamonjisoa, M. and Massa, F., 2025. Dinov3. arXiv preprint arXiv:2508.10104.

---

> > ### Author Rebuttal · Reviewer_gkqp · 2026-04-02
> >
> > The authors' response solves most of my concerns. I will keep my positive score.

---

> > > ### Author Response · Authors · 2026-04-02
> > >
> > > We sincerely thank the reviewer for carefully reading our rebuttal and for confirming that the concerns have been addressed.

---

### Official Review · Reviewer_oPs5 · 2026-03-11

**Soundness:** 3
**Presentation:** 3
**Significance:** 2
**Originality:** 2
**Overall Recommendation:** 4
**Confidence:** 3

**Summary:**

This paper proposes the CoLA framework, which extends LoRA by introducing a dedicated inter-modal adaptation pathway alongside the standard intra-modal one, enabling effective adaptation of unimodal foundation models to multimodal tasks.

**Compliance With Llm Reviewing Policy:**

Affirmed.

**Final Justification:**

The response has addressed most of my key concerns, and I have therefore raised my score to 4.

**Key Questions For Authors:**

See weaknesses

**Limitations:**

Compared to LoRA, CoLA will significantly increase memory usage, and decrease both training and inference speed.

**Strengths And Weaknesses:**

_Strengths_
1. The inter-modal adaptation and intra-modal adaptation design seems to be able to enhance cross-modal interaction in multimodal settings.
2. Experiments show that CoLA works well.

_Weaknessnes_
1. In Table 3, the number of updated parameters for CoLA is significantly higher than that of MaPPER and SwimVG, yet the performance improvement is marginal. Additionally, there is a lack of experiments that either increase the updated parameters of MaPPER and SwimVG to a level comparable to CoLA or reduce CoLA's parameters to match those of MaPPER and SwimVG.
2. The idea of intermediate layer interaction between two encoders does not seem new—it has already appeared in SwimVG [1]. It’s just that CoLA uses LoRA, while SwimVG uses adapters.

[1] SwimVG: Step-wise Multimodal Fusion and Adaption for Visual Grounding

---

> ### Author Rebuttal · Authors · 2026-03-26
>
> Dear Reviewer oPs5,
>
> Thank you for your detailed review and constructive feedback. We address each of your concerns below.
>
>
>
> **W1: Parameter efficiency vs MaPPER and SwimVG** Regarding parameter efficiency and performance comparison with SwimVG and MaPPER:
>
> - **On improvement margin:** The improvement over MaPPER (+4.7% avg on REC) is substantial. For SwimVG, while the REC margin is smaller (+1.6%), CoLA demonstrates its **task-agnostic** adaptation capability by utilizing EEVG's multi-task decoder, CoLA adapts the backbone to perform both REC and RES simultaneously, whereas SwimVG  and MaPPER is designed specifically for REC only, as shown in Table 3. Moreover, CoLA's improvements are **consistent across all benchmarks and evaluation splits**, spanning both vision-language (REC, RES) and audio-visual (AVE, AVS) tasks, a level of generality that SwimVG and MaPPER cannot achieve due to their task-specific and modality-specific designs.
>
>
> - **On parameter comparison:** The higher update ratio in Table 3 is because we apply CoLA on the EEVG architecture, which performs both REC and RES simultaneously. The EEVG multi-task decoder alone accounts for ~9.6% of the 17.2%, with **CoLA's PEFT modules being only ~7.6%**. SwimVG and MaPPER only perform REC and do not require such a decoder. As stated in Section 4.3, Table 3 is intended to compare CoLA applied on the EEVG architecture against existing **task-specific** and **modality-specific** PEFT methods to demonstrate its competitiveness. Matching parameters as suggested would still not provide a controlled comparison, as the differences extend beyond the PEFT modules to the overall architecture, task scope, and decoder design.
>
>
> **W2: Novelty over SwimVG's intermediate-layer interaction** We would like to clarify that the difference between CoLA and SwimVG goes beyond simply replacing adapters with LoRA. While both methods enable cross-modal interaction between dual encoders, they differ in several fundamental aspects:
>
>
> - **Cross-modal information transfer:** SwimVG's cross-modal interaction is **unidirectional** text features are transferred to the vision encoder only. CoLA enables **symmetric bidirectional** interaction, where both modalities simultaneously adapt based on features from the paired modality, as described in Equation 4 and Algorithm 1 of our paper. Furthermore, SwimVG achieves this through module-level insertions (prompts and adapters), while **CoLA operates at the weight level**, directly adapting individual linear components through low-rank decomposition with cross-modal conditioning.
>
>
> - **Task, modality, and architecture generality:** SwimVG inserts task-specific modules designed specifically for vision-language grounding,and has only been demonstrated on a single task (REC), a single modality pair, and a single backbone configuration. CoLA adapts individual linear components through low-rank decomposition with cross-modal conditioning, making it **task-, modality-, and architecture-agnostic**. We demonstrate this across both vision-language (REC, RES) and audio-visual (AVE, AVS) tasks, with diverse backbone architectures including ViT-based (DINOv2) and Swin Transformer-based (SwinV2-L) encoders, without any architectural modification.
>
> To our knowledge, **CoLA is the first method to extend LoRA** with input-conditioned cross-modal adaptation specifically designed for dual-encoder architectures.
>
> We hope this addresses your concerns and are happy to elaborate further.

---

> > ### Author Rebuttal · Reviewer_oPs5 · 2026-04-03
> >
> > Regarding Weakness 1, I believe the authors should provide experiments with comparable parameter counts, as in most cases PEFT performance is positively correlated with the number of updated parameters (before performance saturation).
> >
> > Regarding Weakness 2，could the authors provide unidirectional experiments on CoLA?
> >
> >
> > Furthermore, I am unclear whether the inference speed comparison in Section D. Limitations accounts for merging LoRA weights into the weight matrices. The advantage of LoRA lies not only in parameter-efficient fine-tuning, but also in its ability to merge the LoRA matrices into the original weight matrices, thereby incurring no additional inference latency. I think the authors' baselines should at least include methods that are similarly unable to merge into weight matrices, such as MoE-LoRA series methods, which are also the baselines used in MoKA [1]. Additionally, the deep interaction within the encoder breaks parallelism. If the encoder is large enough to exceed a single GPU's memory, LoRA-based methods can distribute different encodings across different GPUs, whereas CoLA cannot. Therefore, I maintain my score.
> >
> > References
> >
> > [1] MoKA: Multimodal Low-Rank Adaptation for MLLMs. In NeurIPS 2025.
> >
> > --------------------Response to the Second Round of Rebuttal------------------------------
> >
> > Thank you for the authors' efforts. Your response has addressed most of my key concerns, and I have therefore raised my score to 4. I hope the authors will include these experimental results in the final version. It would also be better if results for SwimVG and MaPPER under the same number of updated parameters could be added.

---

> > > ### Author Response · Authors · 2026-04-05
> > >
> > > Dear Reviewer oPs5,
> > >
> > > We thank the reviewer for the continued engagement and thoughtful follow-up. We address each concern below.
> > >
> > > **Comparable parameter counts:**
> > > We agree that PEFT performance is generally correlated with parameter count before saturation, which is why Tables 1 and 2 of our paper already include both rank-matched ($r$=16) and parameter-matched LoRA baselines across all tasks. As shown in those tables, the parameter-matched LoRA baseline already shows signs of saturation across all tasks, while CoLA at the same parameter count consistently outperforms. This directly demonstrates that CoLA's performance gains come from cross-modal adaptation rather than increased parameter count. We summarize the key results here:
> > >
> > > **Vision-Language (REC & RES):**
> > > | Method | Params (M) | REC avg | RES avg |
> > > |--------|-----------|---------|---------|
> > > | LoRA $r$=16 | 28.0 | 82.3 | 72.2 |
> > > | LoRA $r$=54 | 40.6 | 81.8 | 72.3 |
> > > | CoLA $r$=16 | 40.5 | **83.4** | **73.7** |
> > >
> > > **Audio-Visual (AVE & AVS):**
> > > | Method | Params (M) (AVE/AVS) | AVE | AVS |
> > > |--------|-----------|-----|-----|
> > > | LoRA $r$=16 | 6.1 / 28.9 | 79.2 | 80.1 |
> > > | LoRA $r$=54/48 | 18.7 / 44.6 | 79.2 | 80.2 |
> > > | CoLA $r$=16 | 18.6 / 44.8 | **80.7** | **80.9** |
> > >
> > >
> > > **Unidirectional CoLA experiments:**
> > > As requested, we provide additional experiments on AVE comparing unidirectional and bidirectional CoLA in the table below.
> > > | Method | Direction | AVE |
> > > |--------|-----------|-----|
> > > | LoRA | - | 79.2 |
> > > | CoLA | Vision  to Audio only | 79.6 |
> > > | CoLA | Audio to Vision only | 80.2 |
> > > | CoLA | Bidirectional | **80.7** |
> > >
> > > The results show that bidirectional cross-modal adaptation consistently outperforms both unidirectional variants, validating that symmetric cross-modal interaction between both modalities is key to CoLA's performance gains.
> > >
> > > **Comparison with MoE-LoRA:**
> > > We thank the reviewer for suggesting this comparison. We compare against a dense MoE-LoRA baseline at comparable parameter counts in the table below:
> > >
> > > | Method | Params (M) | AVE |
> > > |--------|-----------|-----|
> > > | MoE-LoRA (3 expert) | 16.7 |  79.0 |
> > > | MoE-LoRA (4 expert) | 22.0 | 79.1 |
> > > | CoLA | 18.6 | **80.7** |
> > >
> > > CoLA consistently outperforms MoE-LoRA at comparable parameter counts, demonstrating that cross-modal adaptation rather than input-conditioning drives CoLA's gains.
> > >
> > > **Inference speed:**
> > > We confirm that the inference speed comparison in Figure 6 already accounts for LoRA weight merging. Both LoRA and CoLA's intra-modal pathway weights are fully merged at inference following standard LoRA practices, ensuring a fair comparison. The remaining overhead in CoLA comes solely from the inter-modal pathway, which cannot be merged by design as it relies on dynamic cross-modal features. Importantly, this means that CoLA's **intra-modal pathway parameters are fully eliminated at inference through merging**, unlike LoRA variants such as MoE-LoRA where the input-dependent routing mechanism prevents any weight merging. As shown in Figure 6, CoLA maintains competitive inference speed despite this partial overhead, demonstrating that the inter-modal pathway introduces only modest latency. We will explicitly state the inference merging procedure in the camera-ready version to ensure clarity.
> > >
> > > **GPU parallelism:**
> > > We thank the reviewer for raising this point. When encoders exceed single GPU memory, the common practice in large model PEFT fine-tuning with LoRA, as widely adopted in LLM and VLM fine-tuning, is to use parameter sharding techniques such as FSDP and DeepSpeed ZeRO Stage 3. CoLA is naturally compatible with these approaches, as parameter sharding distributes memory across GPUs while CoLA's cross-encoder interactions remain within the same process without any additional overhead. For tensor parallelism (TP), CoLA's core logic remains unchanged and only requires minor TP-aware implementation of CoLA-modified layers, which is a standard engineering adjustment commonly applied to any custom layer in TP settings. For pipeline parallelism, which presents a more fundamental challenge due to cross-encoder layer dependencies, we note that this challenge is not unique to CoLA as any method that introduces cross-encoder interactions such as SwimVG would face the same constraints. We also note that pipeline parallelism is primarily used in large-scale pretraining rather than PEFT fine-tuning, where FSDP and DeepSpeed ZeRO are the standard parallelism strategies. Supporting pipeline parallelism requires further redesign of CoLA beyond the scope of the current work, and we consider this a promising direction for future work.
> > >
> > > We hope the above clarifications and additional experimental results sufficiently address the reviewer's concerns.

---

### Official Review · Reviewer_11fE · 2026-03-11

**Soundness:** 4
**Presentation:** 3
**Significance:** 3
**Originality:** 3
**Overall Recommendation:** 4
**Confidence:** 3

**Summary:**

This study proposes a dual-stream PEFT method named CoLA, which adds an independent cross-modal low-rank adaptation path to the standard LoRA framework while introducing a hypernetwork to dynamically generate cross-modal fusion weights.
CoLA achieves approximately 3% and 2% improvements over standard LoRA while maintaining low parameter counts, and it is the first to realize multi-task PEFT for visual localization tasks.

**Compliance With Llm Reviewing Policy:**

Affirmed.

**Final Justification:**

it has basically resolved my questions, so I will improving my rating.

**Key Questions For Authors:**

please see weakness

**Limitations:**

yes

**Strengths And Weaknesses:**

Strengths
1. The paper's diagrams are clear and well-drawn, and the dual-path approach is intuitive and easy to understand.
2. CoLA extends the capability of LoRA improving the performance of dual-encoder architectures for multimodal tasks.

Weaknesses
1. The baseline method shows weak performance, and the PEFT method does not appear to incorporate LoRA. Other dual-stream methods such as MMA and MokA were not directly compared in the paper, which reduces the persuasiveness of the results.
2. The improvement achieved by this method over LoRa is marginal, and it comes at the cost of additional parameters and training overhead.
3. While the dual-stream architecture itself has been extensively studied in multimodal literature, CoLA provides a clean LoRA-based realization of intra- and inter-modal adaptation. However, given the large body of prior work (e.g., MMA [CVPR 2024], MokA [NeurIPS 2025]), the novelty of introducing an additional cross-modal low-rank pathway appears somewhat incremental.

---

> ### Author Rebuttal · Authors · 2026-03-26
>
> Dear  Reviewer 11fE,
>
> Thank you for your detailed review and constructive feedback. We address each of your concerns below.
>
> **W1: Missing baselines (MMA, MokA)** We did not include MMA  and MokA  in the main comparison as they target different downstream tasks and type architecture respectively.
>
> - **Comparison with MMA :** MMA  was originally proposed for a different task domain than those we use for evaluation (REC, RES, AVE, and AVS), making it not a directly comparable baseline in our paper. However, to address this concern, we provide additional comparisons of CoLA and MMA on the AVE task, as shown in the table below, where CoLA outperforms MMA by ~2.5%.Both methods use the same backbone and training settings to ensure a fair comparison.
>
> | Method | ACC |
> |--------|-------------|
> | MMA    | 78.18       |
> | CoLA   | **80.7**    |
>
> - **Comparison with MoKA :**  MokA  and CoLA share the same motivation of extending LoRA for multimodal settings but they are applied on completely different architecture types and input configurations. MokA operates within a single shared LLM processing all modality tokens together, while CoLA operates on dual-encoder architectures where each modality has its own separate encoder,  making a direct comparison infeasible. We elaborate on this distinction further in W3.
>
> **W2: Marginal improvement over LoRA with additional overhead**
>
> - **Improvement margin:** CoLA's improvements over LoRA are **consistent across all benchmarks and evaluation splits**, not limited to a single task. Specifically, CoLA outperforms LoRA on all eight evaluation splits of REC (+1.1% avg), all eight splits of RES (+1.5% avg), AVE (+1.5%), and AVS (+0.8%). Furthermore, comparing against the fully fine-tuned EEVG baseline highlights the strength of CoLA's cross-modal adaptation:
>
> | Method | REC avg | RES avg |
> |--------|---------|---------|
> | EEVG (FT) | 81.7 | 72.2 |
> | LoRA | 82.3 (+0.6%) | 72.2 (+0.0%) |
> | CoLA | 83.4 (+1.7%) | 73.7 (+1.5%) |
>
> Notably, LoRA barely improves over the FT baseline on RES (+0.0%), while CoLA achieves +1.5%, demonstrating that cross-modal adaptation is essential for RES. For REC, CoLA achieves nearly 3x the gain of LoRA over the FT baseline (+1.7% vs +0.6%), showing that cross-modal adaptation provides substantially stronger improvements than intra-modal adaptation alone.
>
> - **Additional parameters and overhead:** The parameter-matched comparison is already provided in Tables 1 and 2 of our paper. When LoRA's rank is increased to match CoLA's parameter count (e.g., LoRA r=54 on AVE: 79.2% vs CoLA r=16: 80.7%), **LoRA shows no improvement**, demonstrating that simply adding more parameters does not account for CoLA's gains. The ablation study in Table 5 further confirms this performance improves specifically from **separating intra- and inter-modal pathways**, not from increased parameter count. Regarding computational cost, as shown in Figure 6 (Appendix D), CoLA's GFLOPs are nearly identical to LoRA (5.78 vs 5.82), with only modest increases in memory and training/inference speed a reasonable trade-off given CoLA's consistent gains across all evaluated tasks.
>
> **W3 : Novelty clarification** We want to clarify that CoLA is fundamentally different from both MMA and MokA.
>
> - **MMA:** MMA  is an adapter-based method that uses **static** shared bottleneck weights between modalities, there is no explicit cross-modal interaction, and its behavior is fixed after training regardless of the input. In contrast, CoLA is a **low-rank** adaptation method where the inter-modal pathway generates a **dynamic $\Phi$ matrix** via a hypernetwork, conditioned on features from the paired modality. This means CoLA's cross-modal adaptation is **input-dependent**: each multimodal input pair produces a unique cross-modal interaction, enabling the model to adaptively modulate fusion based on the specific content of both modalities. This is a fundamentally different mechanism from MMA's fixed shared bottleneck or standard LoRA's modality-isolated updates. Furthermore, as shown in the comparison table in W1, our AVE results support this distinction **CoLA (80.7%)** outperforms MMA (78.18%) by a significant margin of **~2.5%**, demonstrating that dynamic, input-conditioned cross-modal adaptation provides a clear advantage over static parameter sharing.
>
> - **MoKA:** As discussed in W1, MoKA targets Multimodal LLMs where all modality tokens are **processed within a single unified architecture** such as LLM, and **cannot be applied to dual-encoder settings**. **CoLA targets dual-encoder architectures** where cross-modal interaction is established between separate encoders, both addressing **multimodal low-rank adaptation but at different architectural levels**. To our knowledge, CoLA is the first method to extend LoRA with input-conditioned cross-modal adaptation specifically designed for dual-encoder architectures.
>
> We hope this addresses your concerns and are happy to elaborate further.

---

> > ### Author Rebuttal · Reviewer_11fE · 2026-04-01
> >
> > Thank you to the authors for their detailed explanation; it has basically resolved my questions, so I will improving my rating.

---

> > > ### Author Response · Authors · 2026-04-01
> > >
> > > Thank you for the acknowledgement. We appreciate your note that our rebuttal has resolved your questions. Given this, we were wondering whether your current rating reflects any remaining substantive concern that has not yet been addressed in the discussion. If so, we would sincerely appreciate a brief indication of that point, and we will do our best to clarify it.

---

### Decision · Program_Chairs · 2026-04-30

**Decision:**

Accept (regular)

**Comment:**

This paper proposed a new PEFT framework called CoLA, to address the modality isolation issue of standard LoRA. Reviewers agree that CoLA can extend the capability of LoRA in the multi-modal setting. During rebuttal, three reviewers' concerns were generally addressed, including concerns on novelty and missing baselines. Finally, two reviewers' opinions were flipped, leading to three out of four reviewers favoring weak accept, which is also my recommendation. However, I urge the authors to include how to handle the $r^2$ scaling concern in their revision.